# When Maximum Entropy Misleads Policy Optimization

Ruipeng Zhang [1]   Ya-Chien Chang [1]   Sicun Gao [1]

## Abstract

The Maximum Entropy Reinforcement Learning (MaxEnt RL) framework is a leading approach for achieving efficient learning and robust performance across many RL tasks. However, MaxEnt methods have also been shown to struggle with performance-critical control problems in practice, where non-MaxEnt algorithms can successfully learn. In this work, we analyze how the trade-off between robustness and optimality affects the performance of MaxEnt algorithms in complex control tasks: while entropy maximization enhances exploration and robustness, it can also mislead policy optimization, leading to failure in tasks that require precise, low-entropy policies. Through experiments on a variety of control problems, we concretely demonstrate this misleading effect. Our analysis leads to better understanding of how to balance reward design and entropy maximization in challenging control problems.

## 1. Introduction

The Maximum Entropy Reinforcement Learning (MaxEnt RL) framework (Ziebart et al., 2008; Abdolmaleki et al., 2018; Haarnoja et al., 2018a; Han & Sung, 2021) augments the standard objective of maximizing return with the additional objective of maximizing policy entropy. MaxEnt methods such as Soft-Actor Critic (SAC) (Haarnoja et al., 2018a) have shown superior performance than other on-policy or off-policy methods (Schulman, 2015; Schulman et al., 2017; Lillicrap, 2015; Fujimoto et al., 2018) in many standard continuous control benchmarks (Achiam, 2018; Raffin et al., 2021; Weng et al., 2021; Huang et al., 2024). Explanations of their performance include better exploration, smoothing of optimization landscape, and enhanced robustness to disturbances (Hazan et al., 2019; Ahmed et al., 2019; Eysenbach & Levine, 2021).

[1]Computer Science and Engineering, UC San Diego. Correspondence to: Ruipeng Zhang <ruz019@ucsd.edu>.

*Proceedings of the $42^{nd}$ International Conference on Machine Learning*, Vancouver, Canada. PMLR 267, 2025. Copyright 2025 by the author(s).

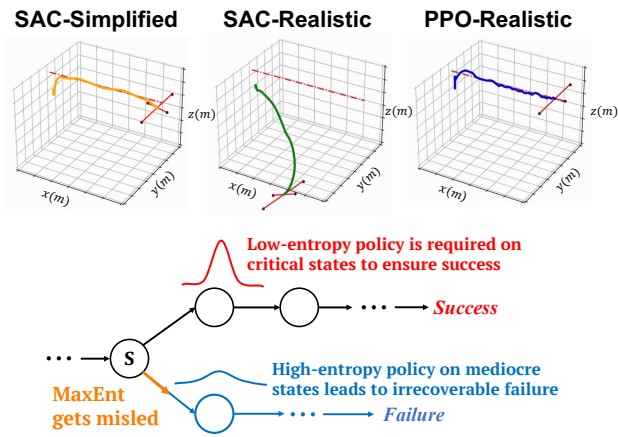

*Figure 1.* (**Upper**) In the quadrotor control environment, SAC learns well under simplified dynamics, but fails to learn when under realistic dynamics models. PPO can learn well despite the use of the latter. (**Lower**) Intuitive illustration of hard control problems, where critical states naturally require low-entropy policies, while MaxEnt RL can favor mediocre states with robust policies of low returns that branch out towards failure and are not recoverable.

Interestingly, the well-motivated benefits of MaxEnt and SAC have not led to its dominance in RL for real-world continuous control problems in practice (Shengren et al., 2022; Tan & Karaköse, 2023; Xu et al., 2021; Radwan et al., 2021). Most recent RL-based robotic control work (Kaufmann et al., 2023; Miki et al., 2022; Zhuang et al., 2024) mostly still uses a combination of imitation learning and fine-tuning with non-MaxEnt methods such as PPO (Schulman et al., 2017). The typical factors of consideration that favor PPO over SAC in practice include computational cost, sensitivity to hyperparameters, and ease of customization. Often the performance by SAC is indeed shown to be inferior to PPO despite efforts in tuning (Muzahid et al., 2021; Lee & Moon, 2021; Nair et al., 2024). In fact, it is easy to reproduce such behaviors. Figure 1 shows the comparison of SAC and PPO for learning to control a quadrotor to follow a path. When the underlying model for the quadrotor is a simplified dynamics model, SAC can quickly learn a stable controller. When a more realistic dynamics model for the quadrotor is used, then SAC always fails, while PPO can succeed under the same initialization and dynamics.

In this paper, we show how the conflict between maximizing

entropy and maximizing overall return can be magnified and hinder learning in performance-critical control problems that naturally require precise, low-entropy policies to solve.

The example of quadcopter control in Figure 1 highlights a common structure in complex control tasks: achieving desired performance often requires executing precise actions at a sequence of critical states. At these states, only a narrow (often zero-dimensional) subset of the action space is feasible, hence the ground-truth optimal policy has inherently low entropy. (In aerodynamic terms, this is often referred to as *flying on the edge of instability*.) Conversely, actions that deviate from this narrow feasible set often lead to states that are not recoverable: once the system enters these states, all available actions tend to produce similarly poor outcomes but accumulate short-term "entropy benefits" that can be favored by MaxEnt. Over time, this drift can compound, ultimately pushing the system into irrecoverable failure.

Consequently, MaxEnt RL may bias the agent toward suboptimal behaviors rather than the precise low-entropy optimal policies that are key to solving hard control problems.

Formalizing this intuition, we give an in-depth analysis of the trade-off. Our main result is that for an arbitrary MDP, there exists an explicit way of introducing entropy traps that we define as *Entropy Bifurcation Extension*, such that MaxEnt methods can be misled to consider an arbitrary policy distribution as MaxEnt-optimal, while the ground-truth optimal policy is not affected by the extension. Importantly, this is not a matter of sample efficiency or exploration bias during training, but is the end result at the convergence of MaxEnt algorithms. Consequently, the misleading effect of entropy can occur easily where standard policy optimization methods are not affected.

We then demonstrate that this concern is not theoretical, and can in fact explain key failures of MaxEnt algorithms in practical control problems. We analyze the behavior of SAC in several realistic control environments, including controlling wheeled vehicles at high speeds, quadrotors for trajectory tracking, quadruped robot control that directly corresponds to hardware platforms. We show that the gap between the value landscapes under MaxEnt and regular policy optimization explains the difficulty of SAC for converging to feasible control policies on these environments.

Our analysis does not imply that MaxEnt is inherently unsuitable for control problems. In fact, following the same analysis, we can now concretely understand why MaxEnt leads to successful learning on certain environments that benefit from robust exploration, including common benchmarking OpenAI Gym environments where SAC generally performs well. Overall, the analysis aims to guide reward design and hyperparameter tuning of MaxEnt algorithms for complex control problems.

We will first give a toy example to showcase the core misleading effect of entropy maximization in Section 4, and then generalize the construction to the technique of entropy bifurcation extension in Section 5. We then show experimental results of how the misleading effects affect learning in practice, and how the adaptive tuning of entropy further validates our analysis in Section 6.

## 2. Related Work

**MaxEnt RL and Analysis.** The MaxEnt RL framework incorporates an entropy term in the RL objective and performs probability matching, such that the policy distribution aligns with the soft-value landscape (Ziebart et al., 2008; Toussaint, 2009; Rawlik et al., 2013; Fox et al., 2015; O'Donoghue et al., 2016; Abdolmaleki et al., 2018; Haarnoja et al., 2018a; Mazoure et al., 2020; Han & Sung, 2021). MaxEnt RL has strong theoretical connections to probabilistic inference (Toussaint, 2009; Rawlik et al., 2013; Levine, 2018) and well-motivated for ensuring robustness from a stochastic inference (Ziebart, 2010; Eysenbach & Levine, 2021) and game-theoretic perspective (Grünwald & Dawid, 2004; Ziebart et al., 2010; Han & Sung, 2021; Kim & Sung, 2023). SAC and algorithms such as SAC-NF (Mazoure et al., 2020), MME (Han & Sung, 2021) and MEow (Chao et al., 2024) have outperformed most non-MaxEnt methods in many standard benchmarking environments (Brockman, 2016; Todorov et al., 2012; Towers et al., 2024). A common explanation of the benefits of MaxEnt is that it enhances exploration (Haarnoja et al., 2018a; Hazan et al., 2019), smoothes the optimization landscape (Ahmed et al., 2019), and solves robust versions of the control problems (Eysenbach & Levine, 2021). Despite the benefits, we show that they can also mislead MaxEnt to converge to suboptimal policies in complex control problems.

**Practical Difficulties with MaxEnt in Control Problems.** RL algorithms are well-known to be sensitive to parameter tuning (Wang & Ni, 2020; Muzahid et al., 2021; Nair et al., 2024). Various recent learning-based robotics control work reported that SAC delivers suboptimal solutions compared to PPO in complex control problems (Tan & Karaköse, 2023; Xu et al., 2021; Radwan et al., 2021). We aim to understand the discrepancy between such results and the generally good performance of MaxEnt algorithms (Haarnoja et al., 2018b; Achiam, 2018; Raffin et al., 2021).

**Trade-off between robustness and optimality.** There is a long line of work studying the trade-off between robustness and performance in supervised deep learning (Su et al., 2018; Zhang et al., 2019; Tsipras et al., 2018; Raghunathan et al., 2019; 2020; Yang et al., 2020). This trade-off in the RL context often overlaps with exploration issues mentioned above. Note that instead of sample efficiency or exploration issues, in this work we focus on pointing out the deeper issue

of misguiding policy optimization results *at convergence*.

## 3. Preliminaries

A Markov Decision Process (MDP) is defined by the tuple: $M = (S, A, P, r, \gamma)$ where: $S$ is the state space, $A$ is the action space, which can be discrete or continuous, $P(s'|s, a)$ is the transition probability distribution, defining the probability of transitioning to state $s'$ after taking action $a$ at state $s$. $r(s, a)$ is the reward function, which specifies the immediate scalar reward received for taking action $a$ at state $s$. $\gamma \in [0, 1)$ is the discount factor. A policy is a mapping $\pi : S \to \Delta A$, where $\Delta A$ is the probability simplex over the action space $A$, defines a probability distribution over actions given any state in $S$. We often write the distribution determined by a policy $\pi$ at a state $s$ as $\pi(\cdot|s)$. The goal of standard RL is to find an optimal policy $\pi^*$ maximizes the expected return over the trajectory of states and actions to achieve the best cumulative reward.

Maximum Entropy RL extends the standard framework by incorporating an entropy term into the objective, encouraging stochasticity in the optimal policy. This modification ensures that the agent not only maximizes reward but also maintains exploration. Instead of maximizing only the expected sum of rewards, the agent maximizes the entropy-augmented objective:

$$J(\pi) = \mathbb{E}[\sum_{t=0}^{\infty} \gamma^t(r(s_t, a_t) + \alpha H(\pi(\cdot|s_t)))] \quad (1)$$

where $H(\pi(\cdot|s)) = -\mathbb{E}_{a \sim \pi(\cdot|s)}[\log \pi(a|s)]$ is the entropy of the policy at state $s$. The coefficient $\alpha \geq 0$ controls weight on entropy. The Bellman backup operator $T^\pi$ is:

$$T^\pi Q(s_t, a_t) = r(s_t, a_t) + \gamma \mathbb{E}_{s_{t+1} \sim p}[V(s_{t+1})], \quad (2)$$

where $V(s_t) = \mathbb{E}_{a_t \sim \pi}[Q(s_t, a_t)] + \alpha H(\pi(\cdot|s_t))$ is the soft state value at $s_t$. Treating Q-values as energy, the Boltzmann distribution induced then at state $s$ is defined as:

$$\pi_Q^*(a|s) = \exp(\alpha^{-1}Q^\pi(s, a))/Z(\pi(\cdot|s)) \quad (3)$$

with $Z(\pi(\cdot|s)) = \int \exp(\alpha^{-1}Q^\pi(s, a))\mathrm{d}a$ as the normalization factor. Policy update in MaxEnt RL at each state $s$ aims to minimize the KL divergence between the policy $\pi(\cdot|s)$ and $\pi_Q^*(\cdot|s)$. Naturally, $\pi = \pi_Q^*$ itself is an optimal policy at state $s$ in the MaxEnt sense, since $D_{\mathrm{KL}}(\pi \| \pi_Q^*) = 0$.

## 4. A Toy Example

From the soft value definitions in MaxEnt, it is reasonable to expect *some* trade-off between return and entropy. But the key to understanding how it can affect the learning *at convergence* in major ways is by introducing intermediate states, where entropy shapes the soft values differently so

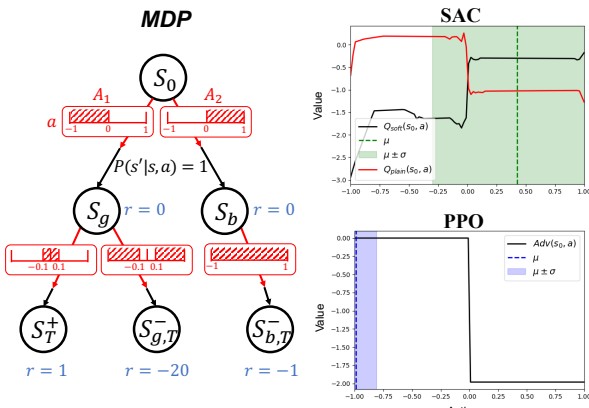

*Figure 2.* (**Left**) MDP in the Toy Example: The MDP consists of an initial state $s_0$ and two subsequent states $s_g$ (good) and $s_b$ (bad). It is clear that an optimal policy for $s_0$ should be centered in the left half of the action interval, since only $s_g$ can transit to the terminal state $s_T^+$ with positive reward. (**Right**) Learning results of SAC and PPO at $s_0$ at convergence. In the SAC plot, the soft Q-values $Q(s_0, a)$ is higher for actions leading to $s_b$, results in an incorrect policy centered in $A_2$, the wrong action region ($\mu$: dashed green line, $\sigma$: green area). We also show the learned Q-values without entropy term with separate networks (red line), showing higher values for actions leading to $s_g$. In the PPO plot, it learns the correct optimal policy.

that the MaxEnt-optimal policy is misled at an *upstream state*. We show a simple example to illustrate this effect. Consider an MDP (depicted in Figure 2) defined as follows:

• State space $S = \{s_0, s_g, s_b, s_T^+, s_{g,T}^-, s_{b,T}^-\}$. Here $s_0$ is the critical state for selecting actions. $s_g$ is the "good" next state for $s_0$, and $s_b$ is the "bad" next state, in the sense that only $s_g$ can transit to the terminal state $s_T^+$ with positive reward (under some subset of actions), while $s_b$ always transits to terminal state $s_{b,T}^-$ with negative reward.

• Action space $A = [-1, 1]$, a continuous interval in $\mathbb{R}$.

• The transitions on $s_0$ are defined as follows, reflecting the intuition mentioned above in the state definition:

– At state $s_0$, any action in $A_1 = [-1, 0)$ leads to the good state deterministically, i.e., $\forall a \in A_1$, $P(s_g|s_0, a) = 1$, while any action in $A_2 = [0, 1]$ leads to the bad state, i.e., $\forall a \in A_2$, $P(s_b|s_0, a) = 1$. Note that allowing zero-dimensional overlap between the $A_1$ and $A_2$ with random transitions at overlapping points will not change the results.

– The transitions from $s_g$: $P(s_T^+|s_g, [-0.1, 0.1]) = 1$ and $P(s_{g,T}^-|s_g, [-1, -0.1) \cup (0.1, 1]) = 1$. That is, for any action in a small fraction of the action space, $a' \in [-0.1, 0.1]$, we can transit to the positive-reward terminal state $s_T^+$.

– The transitions from $s_b$ from any action deterministically lead to the negative-reward terminal state, i.e. $\forall a' \in A, P(s_{b,T}^-|s_b, a') = 1$.

- The rewards are collected only at the terminal states, with $r(s_T^+) = 1$, $r(s_{g,T}^-) = -20$ and $r(s_{b,T}^-) = -1$. The discount factor is set to $\gamma = 0.99$ and $\alpha$ is set to 1.

Given the MDP definition, it is clear that the ground truth policy at $s_0$ should allocate as much probability mass as possible (within the policy class being considered) for actions in the $A_1$ interval, because $A_1$ is the only range of actions that leads to $s_g$, and then has a chance of collecting positive rewards on the terminal state $s_T^+$, if the action on $s_g$ correctly taken.

We can calculate analytically the soft Q values and the policy distributions that are MaxEnt-optimal (shown in Appendix A.1). We can observe that the soft values of $s_g$ and $s_b$ will force MaxEnt to favor $s_b$ at $s_0$. Indeed, as shown in Figure 2, the SAC algorithm with Gaussian policy has its mean converged to the center of $A_2$. The black curve in the SAC plot shows the soft Q-values of the actions, showing how entropy affects the bias. On the other hand, PPO correctly captured the policy that favors the $A_1$ range. More detailed explanations are in Appendix A.1.

We will illustrate how the misleading effect of entropy captured in the toy example can arise in realistic control problems through experimental results in Section 6.

## 5. Entropic Bifurcation Extension

Building on the intuition from the toy example, we introduce a general method for manipulating MaxEnt policy optimization. The method can target arbitrary states in any MDP and "inject" special states with a special configuration of the soft Q-value landscape to mislead the MaxEnt-optimal policy. Importantly, the newly introduced states do not change the optimal policy on any non-targeted original states (in either the MaxEnt or non-MaxEnt sense), but can arbitrarily shape the MaxEnt-optimal policy on the targeted state. Thus, the technique can be applied on any number of states, and in the extreme case to change the entire MaxEnt-optimal policy to mirror the behavior of the worst possible policy on all states, thereby creating an arbitrarily large gap between the MaxEnt-optimal policy and the true optimal policy.

The key to our construction is to introduce new states that create a bifurcating soft Q-value landscape, such that the MaxEnt objective of probability matching biases the policy to favor states with low return and also can not transit back to desired trajectories, thus sabotaging learning.

**Definition 5.1** (Entropic Bifurcation Extension). Consider an arbitrary MDP $M = \langle S_M, A_M, P_M, r_M, \gamma \rangle$ with continuous action space, the entropic bifurcation extension on $M$ at state $s$ is a set $\mathcal{E}(M, s)$ of new MDPs $\hat{M}$ of form:

$$\hat{M} = \langle S_{\hat{M}}, A_{\hat{M}}, P_{\hat{M}}, r_{\hat{M}}, \gamma \rangle \in \mathcal{E}(M, s).$$

constructed with the following steps (illustrated in Figure 3):

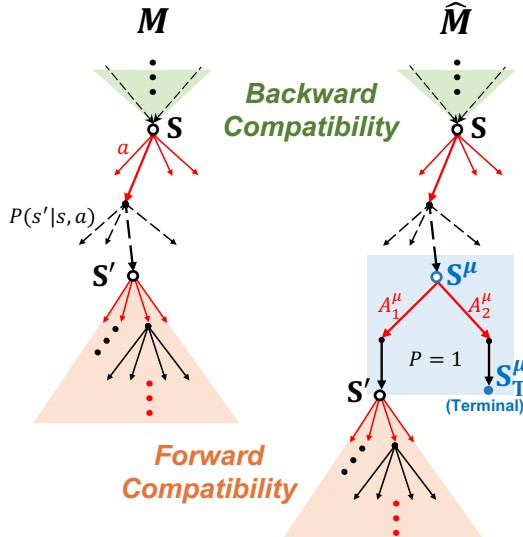

*Figure 3.* MDP $M$ and its entropic bifurcation extension $\hat{M}$. The extension captures the intuition in the toy example, by using additional intermediate states which specifically designed reward to mislead MaxEnt-optimal policies that match the soft-Q landscapes.

1. We write $\mathcal{N}(s) = \{s' | P(s'|s, a) > 0$ for some $a \in A_M\}$ to denote the set of next states with non-zero transition probability from $s$. We introduce new states as follows:

- For any $s' \in \mathcal{N}(s)$, introduce a new state $s^\mu$ that has not appeared in the state space $S_M$ and let the correspondence between $s'$ and $s^\mu$ be marked as $s^\mu = \mu(s')$. We can then write $S^\mu = \mu(\mathcal{N}(s))$ for the set of all new states that are introduced in this way for state $s$.

- At the same time, for each $s^\mu$, we introduce a fresh state $s_T^\mu$ that is a new terminal state, and write the set of such newly introduced terminal states as $S_T^\mu$.

We now let the state space of $\hat{M}$ be $S_{\hat{M}} = S_M \cup S^\mu \cup S_T^\mu$. Note that $S^\mu$, $S_T^\mu$, and $S_M$ are always disjoint.

2. At the original state $s$, for each $s' \in \mathcal{N}(s)$, we now set $P_{\hat{M}}(s'|s, a) = 0$ and $P_{\hat{M}}(\mu(s')|s, a) = P_M(s'|s, a)$. That is, we delay the transition to $s'$ and let the newly introduced state $s^\mu = \mu(s')$ take over the same transitions. For all other states in $S_M \setminus \{s\}$, the transitions in $M$ and $\hat{M}$ are the same.

3. For all the newly introduced states in $S^\mu$, their action space is a new $A^\mu \subseteq \mathbb{R}$. At each $s^\mu \in S^\mu$, we will define transitions on two disjoint intervals, i.e., $A_1^{s^\mu} \cup A_2^{s^\mu} \subseteq A^\mu$ and $A_1^{s^\mu} \cap A_2^{s^\mu} = \emptyset$. This partitioning is $s^\mu$-dependent (they will be used to tune the soft-Q landscapes), but for notational simplicity we will just write $A_1^\mu$ and $A_2^\mu$ when possible. Overall, the new MDP $\hat{M}$ has action space $A_{\hat{M}} = A_M \cup A^\mu$.

4. At each $s^\mu$, the transitions are defined to produce bifurca-

tion behaviors, as follows. For any $a \in A_1^\mu$, $P(s'|s^\mu, a) = 1$. That is, such actions deterministically lead back to the $s'$ in the original MDP. On the other side, any $a \in A_2^\mu$ leads the the new terminal state, $P(s_T^\mu|s^\mu, a) = 1$. That is, the two intervals of action introduce bifurcation into two next states, both deterministically. This design generalizes the construction in the toy example, splitting the action space into one part that recovers the original MDP, and a second part that leads to non-recoverable suboptimal behaviors.

5. The reward function $r_{\hat{M}}$ is the same as $r_M$ on all the original states and actions, i.e., $r_{\hat{M}}(s, a) = r_M(s, a)$ for all $(s, a) \in S_M \times A_M$. The reward on the new state is $r_{\hat{M}}(s^\mu, a) = 0$ for any action $a \in A_2^\mu$. On the new terminal state $s_T^\mu$, we can choose $r(s_T^\mu)$ to shape the soft-Q values as needed. The same discount $\gamma$ is shared between $M$ and $\hat{M}$.

**Notation 5.2.** The construction above defines the set of all possible bifurcation extensions $\mathcal{E}_s(M)$. For any specific instance, the only tunable parameters are the size of $|A_1^\mu|$ and $|A_2^\mu|$, as well as rewards on the newly introduced states $s^\mu$ and $s_T^\mu$. We will show that these parameters already give enough degrees of freedom to shape the soft $Q$-value landscapes and policy at the target state $s$.

Our main theorem relies on two important lemmas, which guarantee that we can use the newly introduced states to arbitrarily shape the policy at the targeted state $s$, without affecting the policy at another state in the original MDP.

- *Backward compatibility* ensures that any fixed value of $V(s)$ can remain invariant, by finding an appropriate soft $Q$-value landscape at $s$. Importantly, this $Q$-landscape can be shaped to match an arbitrary policy distribution at $s$.

- *Forward compatibility* ensures that by choosing appropriate $r(s_\mu^T), |A_1^\mu|, |A_2^\mu|$, the MaxEnt optimal value on $V(s^\mu)$ can match arbitrary target values, using the original values on the original next states $s' \in \mathcal{N}(s)$.

These two properties ensure the feasibility of shaping the MaxEnt-optimal policy at the targeted state $s$ without affecting the policy on any other states of the original MDP. Formally, the lemmas can be stated as follows and the proofs are in the Appendix:

**Lemma 5.3** (Backward Compatibility). *Let $\pi(\cdot|s) : A_M \to [0, 1]$ be an arbitrary policy distribution over the action space at the targeted state $s$. Let $v_s \in \mathbb{R}$ be an arbitrary desired value for state $s$. There exists a value function $V : S^\mu \to \mathbb{R}$ on all the newly introduced states $s^\mu$ such that $v_s$ is the optimal soft value of $s$ under the MaxEnt-optimal policy at $s$ (Definition 4).*

**Lemma 5.4** (Forward Compatibility). *Let $s^\mu = \mu(s')$ be the newly introduced state for $s' \in \mathcal{N}(s)$. Let $V(s')$ be*

an arbitrarily fixed value for the original next state $s'$, and $r(s^\mu, a)$ an arbitrary reward for the newly introduced state $s^\mu$. Let $v \in \mathbb{R}$ be an arbitrary target value. Then, there exist choices of $A_1^\mu$, $A_2^\mu$, and $r(s_T^\mu)$ such that $V(s^\mu) = v$ is the optimal soft value for the bifurcating state $s^\mu$.

The lemmas ensure that for any transition in the original state, $(s, a, s')$, and for any value $V_M^{\pi_M}(s)$ and $V_M^{\pi_M}(s')$ in $M$ under some policy $\pi_M$, there exist parameter choices in the bifurcation extension that maintains the same values of $V_M(s)$ and $V_M(s')$, while shaping the MaxEnt-optimal policy arbitrarily. Consequently, the bifurcation extension can create an arbitrary value gap between the MaxEnt-optimal policy and the ground truth optimal policy at the targeted state $s$. This leads to the main theorem:

**Theorem 5.5** (Bifurcation Extension Misleads MaxEnt RL). *Let $M$ be an MDP with optimal MaxEnt policy $\pi^*$, and $s$ an arbitrary state in $S_M$. Let $\pi(\cdot|s)$ be an arbitrary distribution over the action space $A_M$ at state $s$. We can construct an entropy bifurcation extension $\hat{M}$ of $M$ such that $\hat{M}$ is equivalent to $M$ restricted to $S_M \setminus \{s\}$ and does not change its optimal policy on those states, while the MaxEnt-optimal policy at $s$ after entropy bifurcation extension can follow an arbitrary distribution $\pi(\cdot|s)$ over the actions.*

**Proposition 5.6** (Bifurcation Extension Preserves Optimal Policies). *By setting $r(s^\mu, a) = (1 - \gamma)V(s')$ for every newly introduced bifurcating state $s^\mu = \mu(s')$ and $a \in A_1^\mu$, the optimal policy is preserved under bifurcation extension.*

Now, since this construction can alter the policy at any state without affecting other states, it can be independently used at any number of states simultaneously, and alter the entire policy of the MDP. In particular, the bifurcation extensions can force the MaxEnt-optimal policy to match the worst policy in the original MDP.

**Corollary 5.7.** *Let $M$ be an MDP whose optimal policy has value $J^+$ and its worst policy (minimizing return) has value $J^-$. By applying entropy bifurcation extension on $M$ on all states in $M$, we can obtain an MDP $\hat{M}$ whose MaxEnt-optimal policy has value $J^-$ while its ground-truth optimal policy still has value $J^+$.*

**Remark 5.8.** Our theoretical analysis does not need to use properties of function approximators or other components of practical MaxEnt algorithms, because the MaxEnt-optimal policies can be directly characterized and manipulated, as they must align with the soft-Q landscape. In the next section we show how this theoretical analysis explains the practical behaviors of SAC in realistic control environments.

## 6. Experiments

We now show empirical evidence of how the misleading effect of entropy can play a crucial role in the performance of

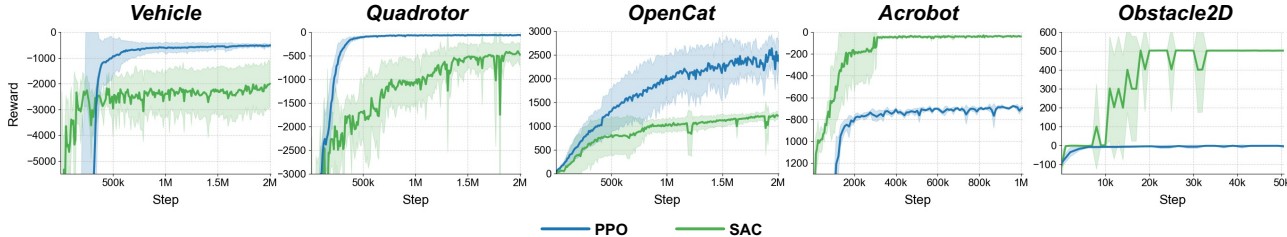

Figure 4. **Reward performance** of SAC and PPO across five environments with five random seeds. Note that we choose to show SAC and PPO because they are the best representatives of MaxEnt and non-MaxEnt algorithms.

MaxEnt algorithms, both when they fail in complex control tasks and when they outperform non-MaxEnt methods.

We first analyze the performance of the algorithms on continuous control environments that involve realistic complex dynamics, including quadrotor control (direct actuation on the propellers), wheeled vehicle path tracking (nonlinear dynamic model at high speed), and quadruped robot control (high-fidelity dynamics simulation from commercial project (PetoiCamp)). We show how the soft value landscapes mislead policy learning at critical states that led to the failure of the control task, while non-MaxEnt algorithms such as PPO can successfully acquire high-return actions.

We also revisit some common benchmark environments to show how the superior performance of MaxEnt can be attributed to the same "misleading" effect that prevents it from getting stuck as non-MaxEnt methods. It reinforces more well-known advantages of MaxEnt with grounded explanations supported by our theoretical understanding.

To further validate our theory, we add new adaptive entropy tuning in SAC, enabling it to switch from soft-Q to normal Q values when their landscapes have much discrepancy. We then observe that the performance of SAC is improved in the environments where it was failing. In particular, the newly learned policy acquires visibly-better control actions on critical states. This form of adaptive entropy tuning is not intended as a new algorithm – it relies on global estimation of Q values that is hard to scale. Instead, the goal is to show the importance of understanding the effect of entropy, as directions for future improvement of MaxEnt algorithms.

**Environments.** In the **Vehicle** environment, the task is to control a wheeled vehicle under the nonlinear dynamic bicycle model (Kong et al., 2015) to move at a constant high speed along a path. Effective control is critical for steering the vehicle onto the path. In the **Quadrotor** environment, the task is to control a quadrotor to track a simple path under small perturbations. The actions are the independent speeds of its four rotors (Rubí et al., 2020), which makes the learning task harder than simpler models. The **Opencat** environment simulates an open-source Arduino-based

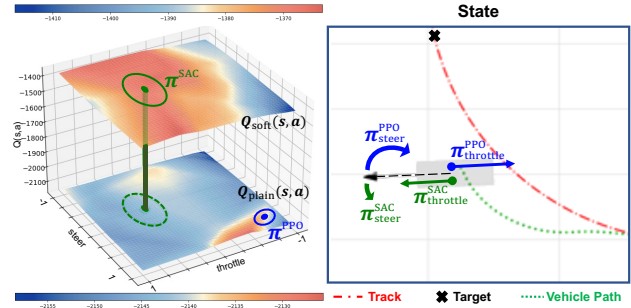

Figure 5. Soft and Plain Q-value landscapes in **Vehicle**. (**Left**) Q landscapes with $Q_{\text{soft}}(s, a)$ and without $Q_{\text{plain}}(s, a)$ entropy. Introduced Entropy in SAC elevates the true Q values to encourage exploration, risking missing the only feasible optimal actions. (**Right**) Rendering of the queried state. The grey rectangle denotes the vehicle with the black arrow as its heading direction. The current SAC policy steers the vehicle left and moves forward, while the PPO policy reasonably steers it back on track with braking, aligning with the optimal region indicated by $Q_{\text{plain}}$.

quadruped robot (PetoiCamp). The action space is the 8 joint angles of the robot. **Acrobot** is a two-link planar robot arm with one end fixed at the shoulder and the only actuated joint at the elbow (Spong, 1995). The action is changed to continuous torque values instead of simpler discrete values in OpenAI Gym (Brockman, 2016). In **Obstacle2D**, the goal is to navigate an agent to the goal behind a wall, which creates a clear suboptimal local policy that the agent should learn to avoid. **Hopper** is the standard MuJoCo environment (Todorov et al., 2012) where SAC typically learns faster and more stably than PPO.

**Overall Performance.** Fig. 4 shows the overall performance comparison of the learning curves of SAC and PPO across environments. We notice that SAC performs worse than PPO in the first three environments that are harder to control under complex dynamics, while significantly outperforming PPO in Acrobot, Obstacle2D, and Hopper (shown in Appendix D.3). Our goal is to understand how these behaviors are affected by entropy in the soft value landscapes.

## 6.1. Misleading Soft-Q Landscapes

In Figure 5, we show the Q values on a critical state in the **Vehicle** control environment, where the vehicle is about to deviate much from the path to be tracked. Because of the high entropy of the policy on the next states where the vehicle further deviates, the soft-Q value landscape at this state is as shown in the top layer on the left in Figure 5. It fails to understand that critical actions are needed, but instead encourages the agent to stay in the generally high soft-Q value region, where the action at the center is shown as the green arrow on the right-hand side plot in the Figure 5. It is clear that the action leads the vehicle to aggressively deviate more from the target path. On the other hand, the plain Q value landscape, as shown in the bottom layer on the left, uses exactly the same states that the SAC agent collected in the buffer to train, and it can realize that only a small region in the action space corresponds to greater plain Q values. The blue dot in the plot, mapped to the blue action vector on the vehicle, shows a correct action direction that steers the vehicle back onto the desired path. Notably, this is the action learned by PPO at this state, and it generally explains the success of PPO in this environment.

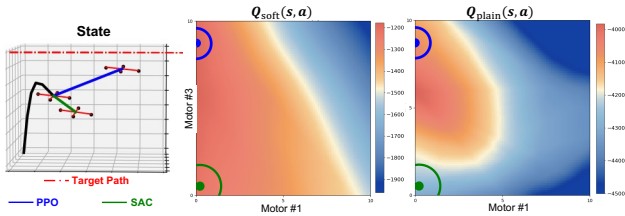

*Figure 6.* Q landscapes in **Quadrotor**. (**Left**) The current state is at the end of the black trajectory. The Red dashed line is the target track. (**Right**) $Q_{\text{soft}}$ and $Q_{\text{plain}}$ at this state. SAC fails to push upward with minimal action at this state, leading to failure against gravity. PPO successfully applies greater thrust to the back motor (#3), flying the quadrotor towards the path.

In Figure 6, we observe similar behaviors in the **Quadrotor** control environment. The quadrotor should track a horizontal path. The controller should apply forward-leaning thrusts to the two rotors parallel to the path, while balancing the other two rotors. The middle and right plots in Figure 6 show the MaxEnt and plain Q values in the action space for the two rotors aligned with the forward direction. Again, because of the high entropy of soft-Q values on mediocre next states that will lead to failure, the value landscape of SAC favors a center at the green dot in the plots, which corresponds to actions of the green arrow in the plot on the left. In contrast, the plain Q value landscape shows that actions of high quality are centered differently. In particular, the blue dot indicates a good action that can be acquired by PPO, controlling the quadrotor towards the right direction.

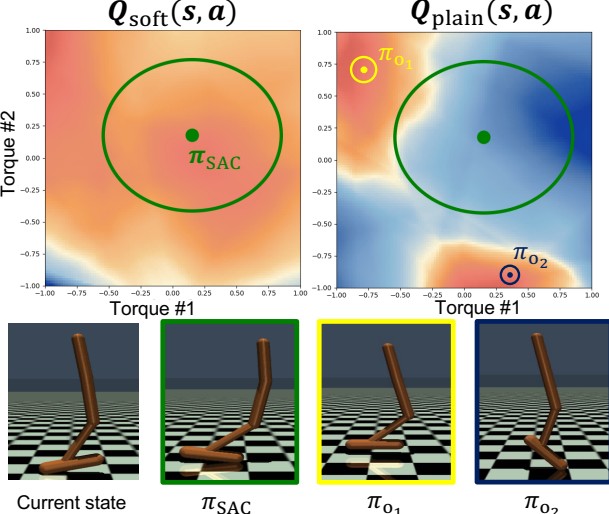

*Figure 7.* (**Upper**) Comparison of soft-Q (left) and plain-Q (right) value landscapes at the current state shown below. (**Lower**) The second to fourth snapshots show Hopper's state after taking actions at the circled position of corresponding colors in the action space shown above. The SAC policy benefits from entropy by 'leaning forward', a risky move despite this action being suboptimal in the current ground truth value.

## 6.2. Benefits of Misleading Landscapes

It is commonly accepted that the main benefit of MaxEnt is the exploratory behavior that it naturally encourages, especially in continuous control problems where exploration is harder to achieve than in discrete action spaces. That is, entropy is designed to "intentionally mislead" to avoid getting stuck at local solutions. We focused on showing how this design can unintentionally cause failure in learning, but the same perspective allows us to more concretely understand the benefits of MaxEnt in control problems. We briefly discuss here and more details are in the Appendix.

Figure 7 shows the comparison between the soft-Q and plain Q landscapes for the training process in **Hopper**, plotted at a particular state shown in the snapshots in the figure. The action learned by SAC is in fact not a high-return action at this point of the learning process. According to the plain Q values, the agent should take actions that lead to more stable poses. However, after this risky move of SAC, the nature of the environment makes it possible to achieve higher rewards, which led to successful learning. Figure 8 shows the similar positive outcome of MaxEnt encouraged by the soft value landscape. Overall, it is important to note that the effectiveness of MaxEnt learning depends crucially on the nature of the underlying environment, and our analysis aims to give a framework for understanding how reward design and entropy-based exploration should be carefully balanced.

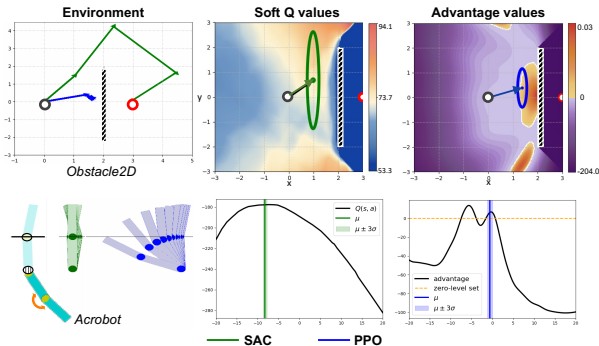

*Figure 8.* Q/Advantage landscapes of SAC and PPO in **Obstacle2D** and **Acrobot**. (**Upper**) In the **Obstacle2D** environment, SAC successfully bypasses the wall while PPO fails, as explained by the Soft-Q/Advantage landscapes. (**Lower**) In **Acrobot**, SAC learns a more stable control policy (applying the right torque to neutralize momentum to prevent failing) while PPO updates are stuck at a local solution that fails to robustly stabilize.

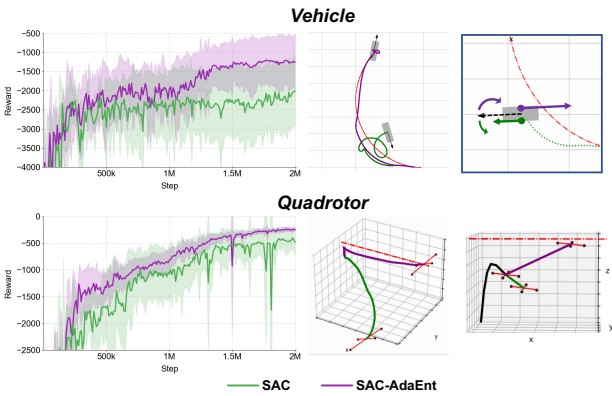

*Figure 9.* Performance of SAC-AdaEnt v.s. SAC. (**Left**) Learning curves. (**Middle**) Full trajectory rendering. (**Right**) Behavior of policy on critic states. In *Vehicle*, SAC-AdaEnt successfully steers and brakes to bring the vehicle back on track. In *Quadrotor*, it effectively lifts the quadrotor to follow the designated path.

### 6.3. Importance of Adaptive Entropy Scaling

To further test the theory that the use of soft values in Max-Ent can negatively affect learning, we modify the SAC algorithm by actively monitoring the discrepancy between the soft Q-value landscape and the plain Q-values.

We simultaneously train two networks for $Q_{\text{soft}}$ and $Q_{\text{plain}}$. During policy updates, we sample from action space at each state under the current policy and evaluate their $Q_{\text{soft}}$ and $Q_{\text{plain}}$ values. If $Q_{\text{soft}}$ deviates significantly from $Q_{\text{plain}}$, it indicates that entropy could mislead the policy, and we rely on $Q_{\text{plain}}$ as the target Q-value for the policy update instead. This adaptive approach, named SAC-AdaEnt, ensures a balance between promoting exploration in less critical states

and prioritizing exploitation in states where the misleading effects may result in failure. Note that SAC-AdaEnt is different from SAC with an auto-tuned entropy coefficient with uniform entropy adjustment across all states.

Figure 9 shows how the adaptive tuning of entropy in SAC-AdaEnt affects learning. In both the **Vehicle** and **Quadrotor** environments, the policy learned by SAC-AdaEnt mostly corrects the behavior of the SAC policy, as illustrated in their overall trajectories and the critical shown in the plots.

Note that the simple change of SAC-MaxEnt is not intended as a new efficient algorithm, because measuring the discrepancy of the Q landscapes requires global understanding at each state, which is unrealistic in high dimensions. It does confirm the misleading effects of entropy in control environments where the MaxEnt approach was originally failing. More details of the algorithm are in Appendix E.

## 7. Conclusion

We analyzed a fundamental trade-off of the MaxEnt RL framework for solving challenging control problems. While entropy maximization improves exploration and robustness, it can also mislead policy optimization towards failure.

We introduced Entropy Bifurcation Extension to show how the ground-truth policy distribution at certain states can become adversarial to the overall learning process in MaxEnt RL. Such effects can naturally occur in real-world control tasks, where states with precise low-entropy policies are essential for success.

Our experiments validated the theoretical analysis in practical environments such as high-speed vehicle control, quadrotor trajectory tracking, and quadruped control. We also showed that adaptive tuning of entropy can alleviate the misleading effects of entropy, but may offset its benefits too. Overall, our analysis provides concrete guidelines for understanding and tuning the trade-off between reward design and entropy maximization in RL for complex control problems. We believe the results also have implications for potential adversarial attacks in RL from human feedback scenarios.

## Acknowledgment

We thank the anonymous reviewers for their helpful comments in revising the paper. This material is based on work supported by NSF Career CCF 2047034, NSF CCF DASS 2217723, and NSF AI Institute CCF 2112665.

## Impact Statement

Our paper offers both theoretical and experimental insights without immediate negative impacts. However, it contributes to a deeper understanding of entropy maximization

principles and their role in reinforcement learning, laying the groundwork for future advancements in MaxEnt RL that may also involve new models of adversarial attacks and defense on RL-based engineering of critical AI systems.

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

# A. Details on the Toy Example

## A.1. Calculation of soft $Q(s_0, a)$ for SAC

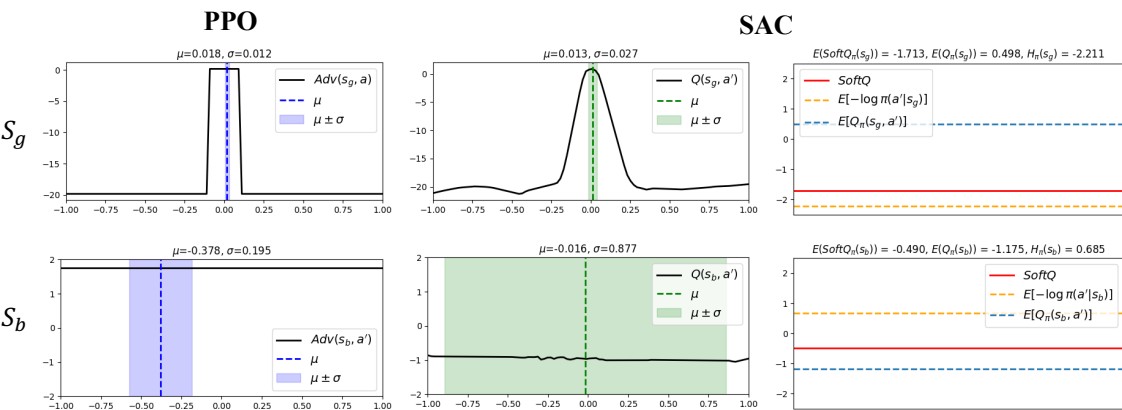

*Figure 10.* Toy Example Results of SAC and PPO at states $s_g$ and $s_b$

In the MaxEnt framework, the policy at $s_0$ is iteratively updated towards the Boltzmann distribution $\pi_Q^*(\cdot|s_0)$. Given the simple transitions in the MDP, we can easily calculate the $Q$ values for any action. We use $\alpha = 1$ for the entropy coefficient.

### A.1.1. DIRECT CALCULATION WITHOUT PARAMETRIZATION

Since transitions from $s_0$ to $s_g$ and $s_b$ are deterministic and yields zero reward, we have

$$Q(s_0, a) = \gamma V(s_{g/b}) = \gamma \mathbb{E}_{a' \sim \pi}[Q(s_{g/b}, a') - \alpha \log \pi(a'|s_{g/b})] = \gamma \int \pi(a'|s_{g/b}) \log Z(s_{g/b}) \mathrm{d}a' = \gamma \log Z(s_{g/b})$$

under the optimal policy $\pi = \pi^* = \exp(\alpha^{-1}Q(s_{g/b}, a'))/Z(s_{g/b})$.

For $a \in [-1, 0)$ which transits to $s_g$,

$$Q(s_0, a) = \gamma \log Z(s_g) = \gamma \log(\int_{-1}^{1} \exp[Q(s_g, a')] \mathrm{d}a') = \gamma \log(\int_{-0.1}^{0.1} e^1 \mathrm{d}a' + \int_{-1}^{-0.1} e^{-20} \mathrm{d}a' + \int_{0.1}^{1} e^{-20} \mathrm{d}a') = -0.603,$$

and $a \in [0, 1]$ which transits to $s_b$,

$$Q(s_0, a) = \gamma \log Z(s_b) = \gamma \log(\int_{-1}^{1} \exp[Q(s_b, a')] \mathrm{d}a') = \gamma \log(\int_{-1}^{1} e^{-1} \mathrm{d}a') = -0.304,$$

As $Q(s_0, [-1, 0)) < Q(s_0, [0, 1])$, SAC is theoretically expected to incorrectly select $a \in [0, 1]$ as the optimal policy.

### A.1.2. CALCULATION WITH EMPIRICAL PARAMETERIZATION

Empirically with Gaussian policy, we can also compute $Q(s_0, a)$ given the policies on $s_g, s_b$, as shown in Figure 10. In practice, $\pi(s_g) = \mathbf{Squash}[\mathcal{N}(\mu(s_g), \sigma(s_g))], \pi(s_b) = \mathbf{Squash}[\mathcal{N}(\mu(s_b), \sigma(s_b))]$, where $\mu(s_g) = 0.013, \sigma(s_g) = 0.027, \mu(s_b) = 0.016, \sigma(s_b) = 0.877$ specifically, thus

$$Q(s_0, a) = \gamma V(s_{g/b}) = \gamma \mathbb{E}_{a' \sim \pi(s_{g/b})}[Q(s_{g/b}, a') - \alpha \log_{\mathcal{N}} \pi(a'|s_{g/b}) + \alpha \log(1 - \tanh^2(a'))]$$

We can compute this numerically as

$$Q(s_0, a) = \gamma V(s_g) \approx -1.696 \ for \ a \in [-1, 0)$$

$$Q(s_0, a) = \gamma V(s_b) \approx -0.485 \ for \ a \in [0, 1]$$

Those values are consistent with the results in Figure 2. Consequently, as expected, MaxEnt algorithms such as SAC quickly converge to the MaxEnt-optimal policy that leads almost all trajectories to the terminal state $s_{b,T}^-$ with negative rewards.

## A.2. Results of PPO policies

The advantage landscapes at $s_0, s_g, s_b$ are shown in Figure 2 and Figure 10. From those, PPO is observed to converge to the correct optimal policy.

## A.3. Remarks on arbitrary $\alpha$

Although we set $\alpha = 1$ in the toy example for simplicity, it can be an arbitrary non-negative value.

**Remark A.1.** If MaxEnt policy is mislead at $s_0$ i.e. $Q(s_0, a|a \in [-1, 0)) < Q(s_0, a|a \in [0, 1])$ when $\alpha = 1$, for arbitrary $\alpha \neq 1$, we can keep misleading MaxEnt policy through reward scaling

$$\hat{r}(s_T^+) = \alpha r(s_T^+), \ \ \hat{r}(s_{g,T}^-) = \alpha r(s_{g,T}^-), \ \ \hat{r}(s_{b,T}^-) = \alpha r(s_{b,T}^-)$$

*Proof.* For arbitrary $\alpha$,

$$Q(s_0, a) = \gamma \mathbb{E}_{a' \sim \pi}[Q(s_{g/b}, a') - \alpha \log \pi(a'|s_{g/b})] = \gamma \alpha \log Z_{Q/\alpha}(s_{g/b})$$

where $\pi$ matches the optimal softmax policy $\frac{\exp(Q(s_{g/b}, a')/\alpha)}{Z_{Q/\alpha}(s_{g/b})}$, and $Z_{Q/\alpha}(s_{g/b}) = \int \exp \frac{Q(s_{g/b}, a')}{\alpha} da'$.

For $\hat{r}(s_T^+), \hat{r}(s_{g,T}^-), \hat{r}(s_{b,T}^-)$, we have $\hat{Q}(s_{g/b}, a') = \alpha Q(s_{g/b}, a')$, and $Z_{Q/\alpha}(s_{g/b}) = \int \exp \hat{Q}(s_{g/b}, a') da$. Therefore the ordering of $Z_{Q/\alpha}(s_g)$ and $Z_{Q/\alpha}(s_b)$ is the same as the original $Z_Q(s_g)$ and $Z_Q(s_b)$. □

**Remark A.2.** For arbitrary $\alpha$, we can always find $r_b^-$ so that the optimal policy for standard RL (e.g. PPO) will favor $s_g$ while MaxEnt policy will favor $s_b$, i.e. $Q(s_0, a|a \in [-1, 0)) < Q(s_0, a|a \in [0, 1])$.

*Proof.* Let $Q(s_0, a|a \in [-1, 0)) < Q(s_0, a|a \in [0, 1])$, we can get

$$\log Z_{Q/\alpha}(s_g) < \log Z_{Q/\alpha}(s_b)$$

$$\log \int \exp \frac{Q(s_g, a')}{\alpha} da' < \log[\exp(\frac{r_b^-}{\alpha}) \cdot |A|]$$

$$\alpha \log \int \exp \frac{Q(s_g, a')}{\alpha} da' - \alpha \log |A| < r_b^- < r^+$$

Given $\alpha \log \int \exp \frac{Q(s_g, a')}{\alpha} da' \leq \max_{a'} Q(s_g, a') + \alpha \log |A|$, we have left-hand side is upper bounded by $\max_{a'} Q(s_g, a') = r^+$. As long as $\sup_{a'} Q(s_g, a') < r^+$, the open interval $\mathcal{I}(s_g) = \left( \alpha \log \int \exp \frac{Q(s_g, a')}{\alpha} da' - \alpha \log |A|, r^+ \right)$ is non-empty. Therefore, we can always pick $r_b^- \in \mathcal{I}(s_g)$ so that $Q(s_0, a|a \in [-1, 0)) < Q(s_0, a|a \in [0, 1])$ i.e. MaxEnt favors $s_b$.

□

# B. Full Proofs

**Lemma B.1** (Lemma 5.3, Backward Compatibility). *Let $\pi(\cdot|s) : A_M \to [0, 1]$ be an arbitrary policy distribution over the action space at the targeted state $s$. Let $v_s \in \mathbb{R}$ be an arbitrary desired value for state $s$. There exists a value function $V : S^\mu \to \mathbb{R}$ on all the newly introduced states $s^\mu$ such that $v_s$ is the optimal soft value of $s$ under the MaxEnt-optimal policy at $s$ (Definition 4).*

*Proof.* Based on the definition of soft value,

$$V(s_t) = \mathbb{E}_{a_t \sim \pi}[Q(s_t, a_t)] + \alpha H(\pi(\cdot|s_t)) \tag{4}$$

we need to show that there exists a function $Q : \{s\} \times A_M \to \mathbb{R}$ such that

$$v_s = V(s) = \mathbb{E}_{a \sim \pi(\cdot|s)}[Q(s, a)] + \alpha H(\pi(\cdot|s))$$

which minimizes the KL-divergence between $\pi$ and the Boltzmann distribution induced by $Q$, i.e.,

$$D_{\mathrm{KL}}(\pi(\cdot \mid s) \parallel \pi_Q^*) = 0.$$

To ensure $D_{\mathrm{KL}}(\pi(\cdot|s)\|\pi_Q^*) = 0$, we can directly construct $Q(s, a)$ such that it matches the optimal policy

$$\pi_Q^*(a|s) = \frac{\exp\left(\alpha^{-1}Q(s,a)\right)}{Z(Q)},$$

with normalization term $Z(Q) = \int_{A_M} \exp\left(\alpha^{-1}Q(s,a)\right)\mathrm{d}a$.

Taking logarithms on both sides and rearranging for $Q(s, a)$:

$$Q(s,a) = \alpha \log \pi(a \mid s) + \alpha \log Z(s), \tag{5}$$

where the normalization factor is a constant that we can arbitrarily choose without changing the KL divergence. Let $c = \alpha \log Z(s)$.

Next, taking expectation over $\pi$:

$$\mathbb{E}_{a\sim\pi(\cdot|s)}[Q(s,a)] = \int_{A_M} \pi(a \mid s)Q(s,a)\mathrm{d}a.$$

Substituting in $Q(s, a)$ from Eq. (5), we have

$$\begin{aligned}
\mathbb{E}_{a\sim\pi(\cdot|s)}[Q(s,a)] &= \int \pi(a|s)\left(\alpha \log \pi(a|s) + c\right)\mathrm{d}a \\
&= \alpha \int \pi(a|s)\log\pi(a|s)\mathrm{d}a + c\int \pi(a|s)\mathrm{d}a \\
&= -\alpha H(\pi(\cdot|s)) + c.
\end{aligned}$$

Now, to match the soft value function $V(s)$, we can set:

$$v_s = \mathbb{E}_{a\sim\pi}[Q(s,a)] + \alpha H(\pi(\cdot|s)) = c - \alpha H(\pi(\cdot|s)) + \alpha H(\pi(\cdot|s)) = c. \tag{6}$$

Thus, solving for $c$, we obtain $c = v_s$.

Substituting back into Eq. (5), we get

$$Q(s,a) = \alpha \log \pi(a \mid s) + v_s.$$

This ensures that both $v_s = V(s) = \mathbb{E}_{a\sim\pi}[Q(s,a)] + \alpha H(\pi(\cdot|s))$ and $D_{\mathrm{KL}}(\pi(\cdot|s)\|\pi_Q^*) = 0$ are satisfied. $\qquad\square$

**Lemma B.2** (Lemma 5.4, Forward Compatibility)**.** *Let $s^\mu = \mu(s')$ be the newly introduced state for $s' \in \mathcal{N}(s)$. Let $V(s')$ be an arbitrarily fixed value for the original next state $s'$, and $r(s^\mu, a)$ an arbitrary reward defined for the newly introduced state $s^\mu$. Let $v \in \mathbb{R}$ be an arbitrarily chosen target value. Then, there exist choices of $A_1^\mu$, $A_2^\mu$, and $r(s_T^\mu)$ such that $V(s^\mu) = v$ is the optimal soft value for the bifurcating state $s^\mu$.*

*Proof.* Following the definition of the MaxEnt value (Definition 4), we need to show:

$$v = V(s^\mu) = \mathbb{E}_{a\sim\pi(\cdot|s^\mu)}[Q(s^\mu, a)] + \alpha H(\pi(\cdot|s^\mu)), \tag{7}$$

where $\pi(\cdot|s^\mu)$ is the policy distribution at $s^\mu$ that exactly matches the Boltzmann distribution induced by some Q-function $Q(s^\mu, a)$, i.e., $D_{\mathrm{KL}}(\pi\|\pi_Q^*) = 0$.

With the bifurcating action space $A^\mu = A_1^\mu \cup A_2^\mu$ and deterministic transitions, we define:

$$Q_1 = r(s^\mu, a) + \gamma V(s'), \tag{8}$$
$$Q_2 = \gamma r(s_T^\mu), \tag{9}$$

where $r(s^\mu, a)$ is an arbitrarily fixed reward for any $a \in A_1^\mu$, and for any $a \in A_2^\mu$, we set $r(s^\mu, a) = 0$. Since $r(s^\mu, a)$ and $V(s')$ are fixed, only $Q_2$ is tunable via the choice of $r(s_T^\mu)$.

A policy that minimizes the KL-divergence with $\pi_Q^*$ at $s^\mu$ is:

$$\pi(a|s^\mu) = \begin{cases} \frac{e^{Q_1/\alpha}}{|A_1^\mu|e^{Q_1/\alpha} + |A_2^\mu|e^{Q_2/\alpha}}, & a \in A_1^\mu, \\ \frac{e^{Q_2/\alpha}}{|A_1^\mu|e^{Q_1/\alpha} + |A_2^\mu|e^{Q_2/\alpha}}, & a \in A_2^\mu. \end{cases}$$

and we define the normalization term as:

$$Z(Q) = |A_1^\mu|e^{Q_1/\alpha} + |A_2^\mu|e^{Q_2/\alpha}.$$

The probabilities over action subspaces are:

$$p_1 = \frac{|A_1^\mu|e^{Q_1/\alpha}}{Z(Q)}, \quad p_2 = \frac{|A_2^\mu|e^{Q_2/\alpha}}{Z(Q)}.$$

The expected value is:

$$\mathbb{E}_{a \sim \pi}[Q(s^\mu, a)] = p_1 Q_1 + p_2 Q_2. \tag{10}$$

The entropy of $\pi(\cdot|s^\mu)$ is:

$$H(\pi(\cdot|s^\mu)) = -(p_1 Q_1 + p_2 Q_2)/\alpha + \log Z(Q),$$

so

$$\alpha H(\pi(\cdot|s^\mu)) = -(p_1 Q_1 + p_2 Q_2) + \alpha \log Z(Q). \tag{11}$$

Substituting Eqs. 10 and 11 into Eq. 7:

$$\begin{aligned} V(s^\mu) &= p_1 Q_1 + p_2 Q_2 + \alpha H(\pi(\cdot|s^\mu)) \\ &= p_1 Q_1 + p_2 Q_2 - (p_1 Q_1 + p_2 Q_2) + \alpha \log Z(Q) \\ &= \alpha \log Z(Q). \end{aligned} \tag{12}$$

Thus, in this corrected version, we observe that:

$$V(s^\mu) = \alpha \log \left( |A_1^\mu|e^{Q_1/\alpha} + |A_2^\mu|e^{Q_2/\alpha} \right).$$

Solving for $r(s_T^\mu)$, we get:

$$r(s_T^\mu) = \frac{1}{\gamma} \left[ \alpha \log \left( \frac{e^{v/\alpha} - |A_1^\mu|e^{Q_1/\alpha}}{|A_2^\mu|} \right) \right].$$

which is valid for all $\alpha > 0$, and the function

$$V(s^\mu) = \alpha \log Z(Q)$$

is a surjection onto $\mathbb{R}$ when varying over $|A_1^\mu|$, $|A_2^\mu|$, and $r(s_T^\mu)$, we conclude that for any $v \in \mathbb{R}$, a valid construction exists such that $V(s^\mu) = v$. $\qquad \square$

**Theorem B.3** (Theorem 5.5, Bifurcation Extension Misleads MaxEnt RL). *Let $M$ be an MDP with optimal MaxEnt policy $\pi^*$, and $s$ an arbitrary state in $S_M$. Let $\pi(\cdot|s)$ be an arbitrary distribution over the action space $A_M$ at state $s$. We can construct an entropy bifurcation extension $\hat{M}$ of $M$ such that $\hat{M}$ is equivalent to $M$ restricted to $S_M \setminus \{s\}$ and does not change its optimal policy on those states, while the MaxEnt-optimal policy at $s$ after entropy bifurcation extension can follow an arbitrary distribution $\pi(\cdot|s)$ over the actions.*

In other words, without affecting the rest of the MDP, we can introduce bifurcation extension at an arbitrary state such that the MaxEnt optimal policy becomes arbitrarily bad at the affected state.

*Proof.* Following Lemma 5.4, we introduce the bifurcation extension as Definition 5.1 and obtain Q-values on all the newly introduced states $Q(s, a)$ such that $V(s)$ remains unchanged, while the MaxEnt optimal policy at $s$ becomes $\pi(\cdot|s)$. Given such target $Q(s, a)$, which now impose target values on the introduced bifurcation states, i.e., $V(s^\mu) = Q(s, a)/P(s'|s, a)$, because by construction $P(s^\mu|s, a) = P(s'|s, a) > 0$. We then use the forward compatibility Lemma 5.4 to set the parameters in the bifurcation extension, such that $V(s^\mu)$ is attained by the MaxEnt policy at $s^\mu$, without changing the existing values on the original next states $V(s')$ for any $s' \in \mathcal{N}(s)$. Since we have not changed the values on $s$ or any $s' \in \mathcal{N}(s)$, the bifurcation extension does not affect the policy on any other state in $S_M \setminus \{s\}$. At the same time, the target arbitrary policy $\pi(\cdot|s)$ is now a MaxEnt optimal policy at $s$. $\qquad\square$

**Proposition B.4** (Proposition 5.6, Bifurcation Extension Preserves Optimal Policies). *By setting $r(s^\mu, a) = (1 - \gamma)V(s')$ for every newly introduced bifurcating state $s^\mu = \mu(s')$ and $a \in A_1^\mu$, the optimal policy is preserved under bifurcation extension.*

*Proof.* In the non-MaxEnt setting, the state value of $s^\mu$ maximizes the $Q$-value, and the optimal policy chooses the actions in $A_1^\mu$. Since $r(s^\mu, a) = (1 - \gamma)V(s')$, the additional reward on $(s^\mu, a)$ ensures that $V(s^\mu) = V(s')$. Note that Lemma 5.4 holds for arbitrary $r(s^\mu, a)$. Consequently, there is no change in $Q(s, a)$ and the optimal policy remains the same between $M$ and $\hat{M}$. $\qquad\square$

# C. Environments

## C.1. Vehicle

The task is to control a wheeled vehicle to maintain a constant high speed while following a designated path. Practically, the vehicle chases a moving goal, which travels at a constant speed, by giving negative rewards to penalize the distance difference. Also the vehicle receives a penalty for deviating from the track. The overall reward is

$$r = r_{\text{goal}} + r_{\text{track}} = -||p_{\text{vehicle}} - p_{\text{goal}}||_2 + \beta_b|R_{\text{vehicle}}^2 - R_{\text{track}}^2|$$

where $p_{\text{vehicle}}, p_{\text{goal}}$ are the positions for vehicle and the goal respectively, $\beta_b = 0.3$ is the scaling factor, $R_{\text{track}} = 10$ is the radius of the quarter-circle track and $R_{\text{vehicle}} = \sqrt{x_{\text{vehicle}}^2 + y_{\text{vehicle}}^2}$ is the vehicle's radial distance from the origin. The initial state is set to make steering critical for aligning the vehicle with the path, given an initial forward speed of $v = 3$. The action space is steering and throttle. The vehicle follows a dynamic bicycle model (Kong et al., 2015), where throttle and steering affect speed, direction, and lateral dynamics. It introduces slip, acceleration, and braking, requiring the agent to manage stability and traction for precise path tracking.

## C.2. Quadrotor

The task is to control a quadrotor to track a simple path while handling small initial perturbations. The quadrotor also chases a target moving at a constant speed. The reward is given by the distance between the quadrotor and the target, combined with a penalty on the quadrotor's three Euler angles to encourage stable orientation and prevent excessive tilting.

The overall reward function is:

$$r = r_{\text{goal}} + r_{\text{stability}} = -\beta||p_{\text{quadrotor}} - p_{\text{target}}||_2 - |\theta| - |\phi| - |\psi|$$

where $p_{\text{quadrotor}}, p_{\text{target}}$ are the positions for quadrotor and the target respectively, $\theta, \phi, \psi$ are the Euler angles, $\beta = 5$ is a scaling factor. **Simplified:** Since the track aligns with one of the rotor axes, we fix the thrust of the orthogonal rotors to zero, providing additional stabilization. The agent controls only the thrust and pitch torque, simplifying the task. **Realistic:** The agent must fully control all four rotors, with the action space consisting of four independent rotor speeds, making stabilization and trajectory tracking more challenging.

## C.3. Opencat

The Opencat environment simulates an open-source Arduino-based quadruped robot, which is based on Petoi's OpenCat project (PetoiCamp). The task focuses on controlling the quadruped's joint torques to achieve stable locomotion while adapting to perturbations. The action space consists of 8 continuous joint torques, corresponding to the two actuated joints for each of the four legs. The agent must learn to coordinate leg movements efficiently to maintain balance and move toward.

## C.4. Acrobot

Acrobot is a two-link planar robot arm with one end fixed at the shoulder ($\theta_1$) and an actuated joint at the elbow ($\theta_2$) (Spong, 1995). The control action for this underactuated system involves applying continuous torque at the free joint to swing the arm to the upright position and stabilize it. The task is to minimize the deviation between the joint angle ($\theta_1$) and the target upright position ($\theta_1 = \pi$), while maintaining zero angular velocity when the arm is upright. The reward function is defined as follows:

$$r = -\left( (\theta_1 - \pi)^2 + (\theta_2)^2 + 0.1(\dot{\theta}_1)^2 + 0.1(\dot{\theta}_2)^2 \right)$$

## C.5. Obstacle2D

The goal is to navigate an 2D agent to the goal $(3,0)$ while avoiding a wall spanning $y = [-2, 2]$ starting from $(0,0)$. The action range is $[-3, 3]$, which makes it sufficient for the agent to avoid the wall in one step. The reward function is based on progress toward the goal, measured as the difference in distance before and after taking a step. For special cases, it receives +500 for reaching the goal, -200 for hitting the wall.

## C.6. Hopper

Hopper is from OpenAI Gym based on mujoco engine, which aims to hop forward by applying action torques on the joints.

# D. Details on Experiments

## D.1. Training without entropy in target Q values

In Sec. 4 and Sec. 6, we simultaneously train Q networks with (soft) and without (plain) the entropy term in the target Q values, in order to illustrate the effect of entropy on policy optimization. Specifically,

$$\mathcal{T}^\pi Q_{\text{soft}}(s_t, a_t) = r(s_t, a_t) + \gamma \mathbb{E}_{s_{t+1}, a_{t+1}}[Q_{\text{soft}}(s_{t+1}, a_{t+1}) - \alpha \log \pi(a_{t+1}|s_{t+1})]]$$
$$\mathcal{T}^\pi Q_{\text{plain}}(s_t, a_t) = r(s_t, a_t) + \gamma \mathbb{E}_{s_{t+1}, a_{t+1}}[Q_{\text{plain}}(s_{t+1}, a_{t+1})]]$$

with the rest of the SAC algorithm unchanged. We still update the policy based on the target Q with entropy, i.e. $Q_{\text{soft}}(s_t, a_t)$ as original SAC and training $Q_{\text{plain}}(s_t, a_t)$ is just for better understanding for entropy's role in the policy updating dynamics.

## D.2. Experiment Hyperparameters

The hyperparameters for training the algorithms are in Table 1 and Table 2.

## D.3. Performance of DDPG and SAC with Auto-tuned Entropy Coefficient

We also run SAC with auto-tuned $\alpha$ and DDPG across all six environments, as shown in Fig. 11. The first row includes environments where SAC fails due to critical control requirements, while the second row shows cases where SAC performs better. Notably, auto-tuning the entropy temperature in SAC improves performance in some critical environments but not all, and it still fails to surpass PPO.

## D.4. Benefits of Misleading Landscapes in SAC

Nonetheless, entropy in target Q is beneficial as designed because of necessary exploration. In Gym *Hopper*, we investigate the state shown in Fig. 12. Entropy smooths the Q landscape in regions that may not produce optimal actions at the current training stage, encouraging exploration and enabling the policy to achieve robustness rather than clinging solely to the current optima.

## D.5. PPO Trapped by Advantage Zero-Level Sets.

Without extra entropy to encourage exploration, PPO as an on-policy RL algorithm can be trapped in the zero-level set of advantages. In Obstacle2D (Fig. 13 first row), we plot the policy of the initial state, where the optimal action is to move to either the upper or lower corner of the wall, avoiding it in one step. The reward is designed to encourage the agent to approach the goal while penalizing collisions with a large negative reward. The region in front of the wall is a higher-reward

*Table 1.* Hyperparameters for SAC(SAC-auto-alpha), PPO, and DDPG

| Hyperparameter | SAC / SAC-auto-$\alpha$ | PPO | DDPG |
|---|---|---|---|
| Discount factor ($\gamma$) | 0.99 | 0.99 | 0.99 |
| Entropy coefficient ($\alpha$) | 0.2/ N/A | 0 | / |
| Exploration noise | 0 | 0 | 0.1 |
| Target smoothing coefficient ($\tau$) | 0.005 | / | 0.005 |
| Batch size | 256 | 64 | 256 |
| Replay buffer size | 1M | 2048 | 1M |
| Hidden layers | 2 | 2 | 2 |
| Hidden units per layer | 256(64 for Toy Example) | 256(64 for Toy Example) | 256 |
| Activation function | ReLU | Tanh | ReLU |
| Optimizer | Adam | Adam | Adam |
| Number of updates per environment step | 1 | 10 | 1 |
| Clipping parameter ($\epsilon$) | / | 0.2 | / |
| GAE parameter ($\lambda$) | / | 0.95 | / |

*Table 2.* Learning Rates for SAC and PPO Across Different Environments

| Algorithm | Learning Rate | Vehicle | Quadrotor | Opencat | Acrobot | Obstacle2D | Hopper |
|---|---|---|---|---|---|---|---|
| SAC | Actor | 1e-3 | 3e-4 | 1e-3 | 1e-3 | 1e-3 | 1e-3 |
| | Q-function | 1e-3 | 3e-4 | 1e-3 | 1e-3 | 1e-3 | 1e-3 |
| SAC auto-$\alpha$ | $\alpha$ | 1e-3 | 3e-4 | 1e-3 | 1e-3 | 1e-3 | 1e-3 |
| PPO | Actor | 3e-4 | 3e-4 | 1e-4 | 3e-4 | 3e-4 | 3e-4 |
| | Value function | 3e-4 | 3e-4 | 1e-4 | 3e-4 | 3e-4 | 3e-4 |
| DDPG | Actor | 1e-3 | 3e-4 | 3e-4 | 1e-3 | 1e-3 | 1e-3 |
| | Q-function | 1e-3 | 3e-4 | 3e-4 | 1e-3 | 1e-3 | 1e-3 |

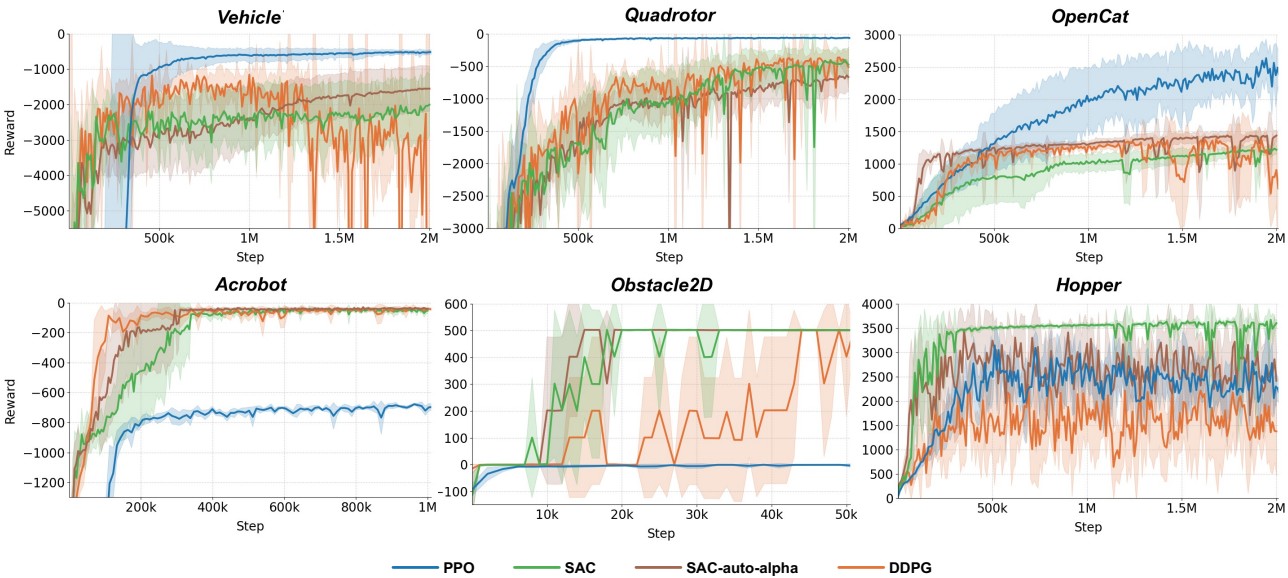

*Figure 11.* Performance of All Algorithms across six environments

area because of the instant approaching reward. The advantage landscape reveals that PPO's policy moves to the center of the positive advantage region but remains confined by the zero-level set. Notably, although the optimal regions (upper and lower corners) have positive advantages, PPO remains trapped due to its local behavior. The coupling of exploration and actual policy worsens this issue—if PPO fails to explore actions to bypass the wall, its policy's $\sigma$ shrinks, further reducing

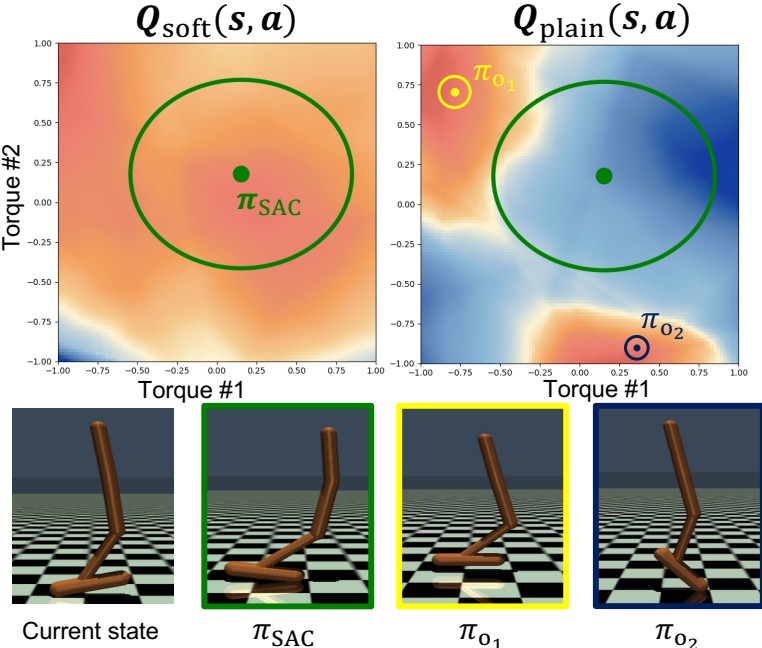

*Figure 12.* **Q landscapes in *Hopper*. Upper**: We set torque #0 (top torso) as the current $\mu_0^{SAC}$ to plot $Q_{\text{soft}}$ and $Q_{\text{plain}}$ for torque #1 (middle thigh) and #2 (bottom leg) in the state shown in the bottom-left figure. **Lower**: Rendered hopper's gestures result from the corresponding policies. SAC's policy benefits from entropy by 'leaning further forward', taking a risky move despite this action being suboptimal in the current true Q. Investigating the peaks $o_1$ and $o_2$ in $Q_{\text{plain}}$ reveals that the hopper tends to 'bend its knee' and 'jump up' when following the corresponding policies, demonstrating less exploration.

exploration and leading to entrapment. We can also observe this across training stages, as shown in Fig. 14.

A similar phenomenon is observed in Acrobot (Fig. 13 second row), where PPO's policy shrinks prematurely, leading to insufficient exploration.

However, this phenomenon can also be viewed as a strength of PPO, as it builds on the current optimal policy and makes incremental improvements step by step, thus not misled by suboptimal actions introduced by entropy. As a result, PPO performs better in environments where the feasible action regions are small and narrow in the action space, such as in *Vehicle*, *Quadrotor*, and *OpenCat*, which closely resemble real-world control settings.

## E. Details on SAC-AdaEnt

### E.1. Pseudocode

We provide the detailed algorithm in Algorithm 1.

### E.2. SAC-AdaEnt improves performance in environments that SAC fails

To further validate the misleading entropy claim and enhance SAC's performance in critical environments, we propose SAC with Adaptive Entropy (SAC-AdaEnt) and test it on *Vehicle* and *Quadrotor* environments, showing improvements in Fig. 15. In these environments, SAC relies excessively on entropy as it dominates the soft Q values. To address this, SAC-AdaEnt adaptively combines target Q values with and without entropy. Specifically, we simultaneously train $Q_{\text{soft}}$ and $Q_{\text{plain}}$ as in Appendix D.1. During policy updates, we sample multiple actions per state under the current policy and evaluate their $Q_{\text{soft}}$ and $Q_{\text{plain}}$ values. By comparing these values, we compute the similarity of the two landscapes. If $Q_{\text{soft}}$ deviates significantly from $Q_{\text{plain}}$, indicating that entropy could mislead the policy, we rely on $Q_{\text{plain}}$ as the target Q value instead. Otherwise, entropy is retained to encourage exploration. This adaptive approach ensures a balance between safe exploration and exploitation, promoting exploration in less critical states and prioritizing exploitation in states where errors could result in failure. Note that SAC-AdaEnt is fundamentally different from SAC with an auto-tuned entropy coefficient,

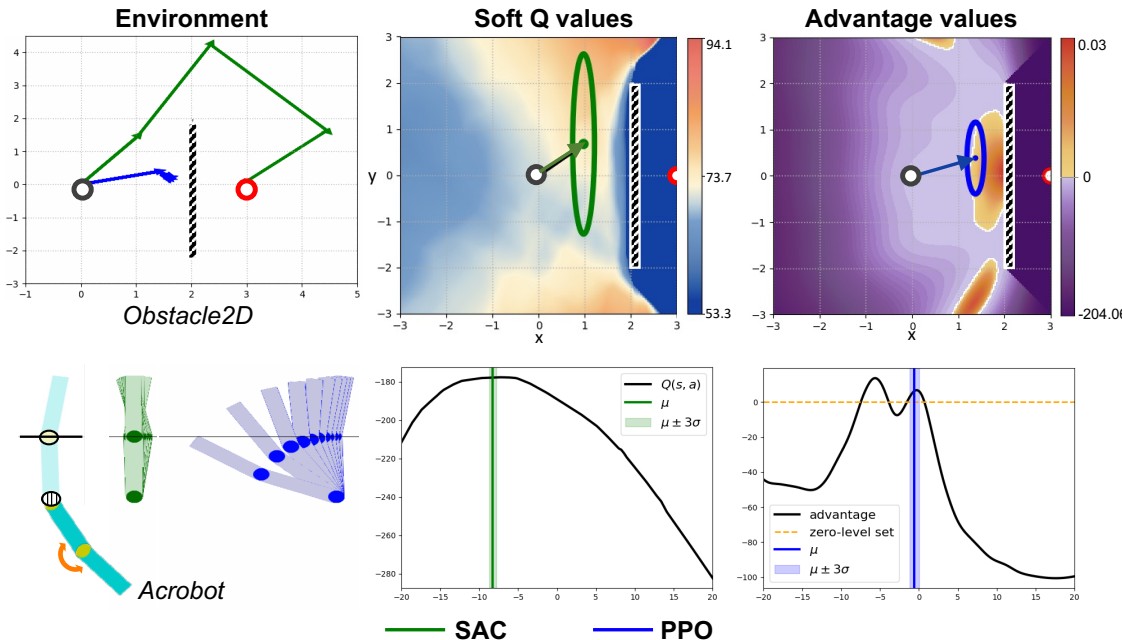

*Figure 13.* **Q/Advantage landscapes** of SAC and PPO in ***Obstacle2D*** and ***Acrobot***. **Upper:** In *Obstacle2D* with start $(0,0)$, goal $(3,0)$, and a wall at $x = 2$ spanning $y = [-2, 2]$, SAC succeeds in bypassing the wall whereas PPO fails. We plot the Q/Advantage landscape of the initial state. For SAC, entropy encourages exploration, guiding updates toward the upper and lower ends of the wall via soft Q. In contrast, PPO remains trapped despite the presence of positive advantage regions near the wall's ends. **Lower:** In *Acrobot*, both algorithms achieve swing-up, but near the stabilization height, SAC applies the right torque to neutralize momentum, preventing it from falling again. In contrast, PPO remains stuck in a local optimum, leading to repeated failures.

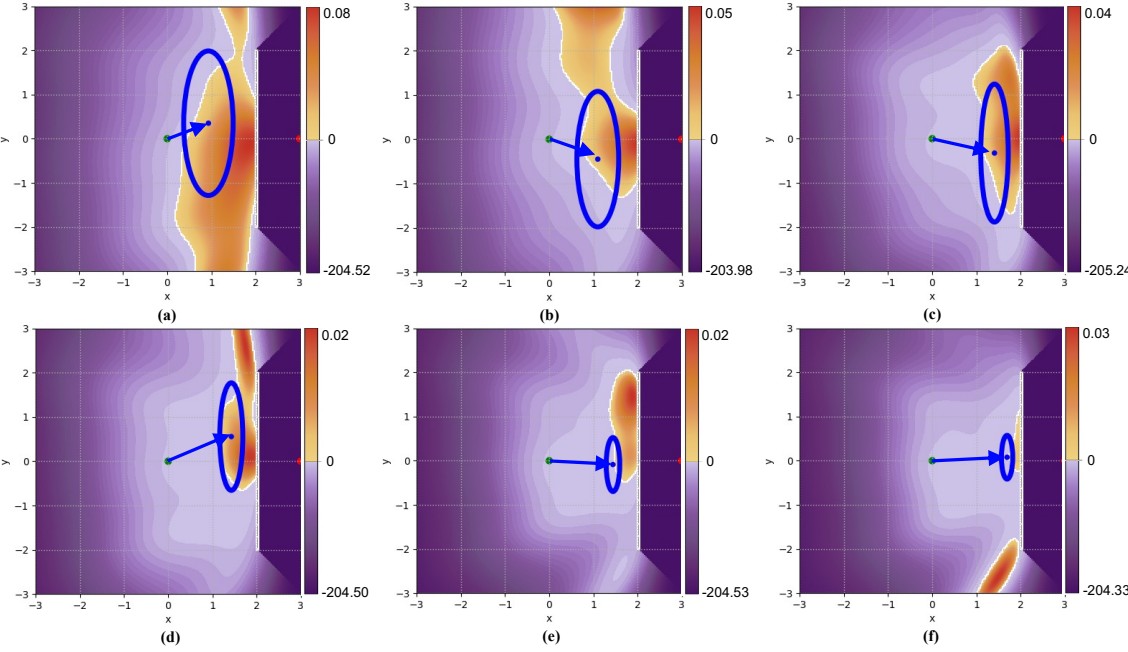

*Figure 14.* Advantage landscapes in *Obstacle2D* for PPO. (a) to (f) show the advantage landscapes at different training stages for the initial state, where PPO's policy center remains trapped in front of the wall while $\sigma$ gradually shrinks.

---

**Algorithm 1** SAC with Adaptive Entropy (SAC-AdaEnt)

---

**Initialize:** Actor network $\pi_\theta$, Q networks and their paired target networks for Q-target with/without entropy $\phi_1, \phi_2, \phi_{targ,1}, \phi_{targ,2}$ (w/ entropy), $\phi'_1, \phi'_2, \phi'_{targ,1}, \phi'_{targ,2}$ (w/o entropy), replay buffer $\mathcal{D}$, similarity threshold $\epsilon$

**for** each training step **do**

    Sample action $a_t \sim \pi_\theta(a_t|s_t)$ and observe $s_{t+1}, r_t$

    Store $(s_t, a_t, r_t, s_{t+1})$ in replay buffer $\mathcal{D}$

    **for** each gradient update step **do**

        Sample minibatch of transitions $(s, a, r, s')$ from $\mathcal{D}$

        Compute target value:

$$y = r + \gamma \left( \min_{i=1,2} \hat{Q}_{\phi_i}(s', a') - \alpha \log \pi_\theta(a'|s') \right), \ y' = r + \gamma \left( \min_{i=1,2} \hat{Q}_{\phi'_i}(s', a') \right)$$

        Update Q networks:

$$\phi_i \leftarrow \phi_i - \eta_Q \nabla_{\phi_i} \frac{1}{N} \sum_{n=1}^N (\phi_i(s, a) - y)^2, \ \phi'_i \leftarrow \phi'_i - \eta_Q \nabla_{\phi'_i} \frac{1}{N} \sum_{n=1}^N (\phi'_i(s, a) - y')^2$$

        For each $s$, sample actions using current policy $A_s = \{a_s | a_s \sim \pi_\theta(\cdot|s)\}$

        Compute similarity score:

$$\text{sim}(\mathbf{Q}, \mathbf{Q}') = \frac{\mathbf{Q}(s) \cdot \mathbf{Q}'(s)}{\|\mathbf{Q}(s)\|\|\mathbf{Q}'(s)\|}, \text{where } \mathbf{Q}(s) = \left[ \min_{i=1,2} \hat{Q}_{\phi_i}(s, a_s) \right]_{a_s \sim \pi_\theta(a|s)}, \ \mathbf{Q}'(s) = \left[ \min_{i=1,2} \hat{Q}_{\phi'_i}(s, a_s) \right]_{a_s \sim \pi_\theta(a|s)}$$

        Update actor policy using reparameterization trick:

$$\theta \leftarrow \theta - \eta_\pi \nabla_\theta \mathbb{E}_{s \sim \mathcal{D}, a \sim \pi_\theta} \begin{cases} \alpha \log \pi_\theta(a|s) - Q_{\phi_1}(s, a), & \text{if } \text{sim}(\mathbf{Q}, \mathbf{Q}') > \epsilon \\ \alpha \log \pi_\theta(a|s) - Q_{\phi'_1}(s, a), & \text{otherwise} \end{cases}$$

        Update target networks:

$$\hat{Q}_{\phi_i} \leftarrow \tau Q_{\phi_i} + (1 - \tau)\hat{Q}_{\phi_i}, \ \hat{Q}_{\phi'_i} \leftarrow \tau Q_{\phi'_i} + (1 - \tau)\hat{Q}_{\phi'_i}$$

    **end for**

  **end for**

---

which applies a uniform entropy adjustment across all states. In contrast, SAC-AdaEnt adaptively adjusts entropy for each state, making it particularly effective in environments requiring precise control and careful exploration.

### E.3. SAC-AdaEnt preserves performance in environments that SAC succeeds

Not only SAC-AdaEnt improves performance in environments where SAC struggles, but it also retains SAC's strengths in those where SAC already excels. We report SAC-AdaEnt's results on *Hopper*, *Obstacle2D*, and *Acrobot* as in Table. 3:

| Algorithm | Hopper | Obstacle2D | Acrobot |
|---|---|---|---|
| SAC | $3484.46 \pm 323.87$ | $501.98 \pm 0.62$ | $-45.25 \pm 7.94$ |
| SAC-AdaEnt | $3285.17 \pm 958.43$ | $501.50 \pm 0.57$ | $-36.31 \pm 16.42$ |

*Table 3.* Performance (mean $\pm$ std) of SAC and SAC-AdaEnt across tasks.

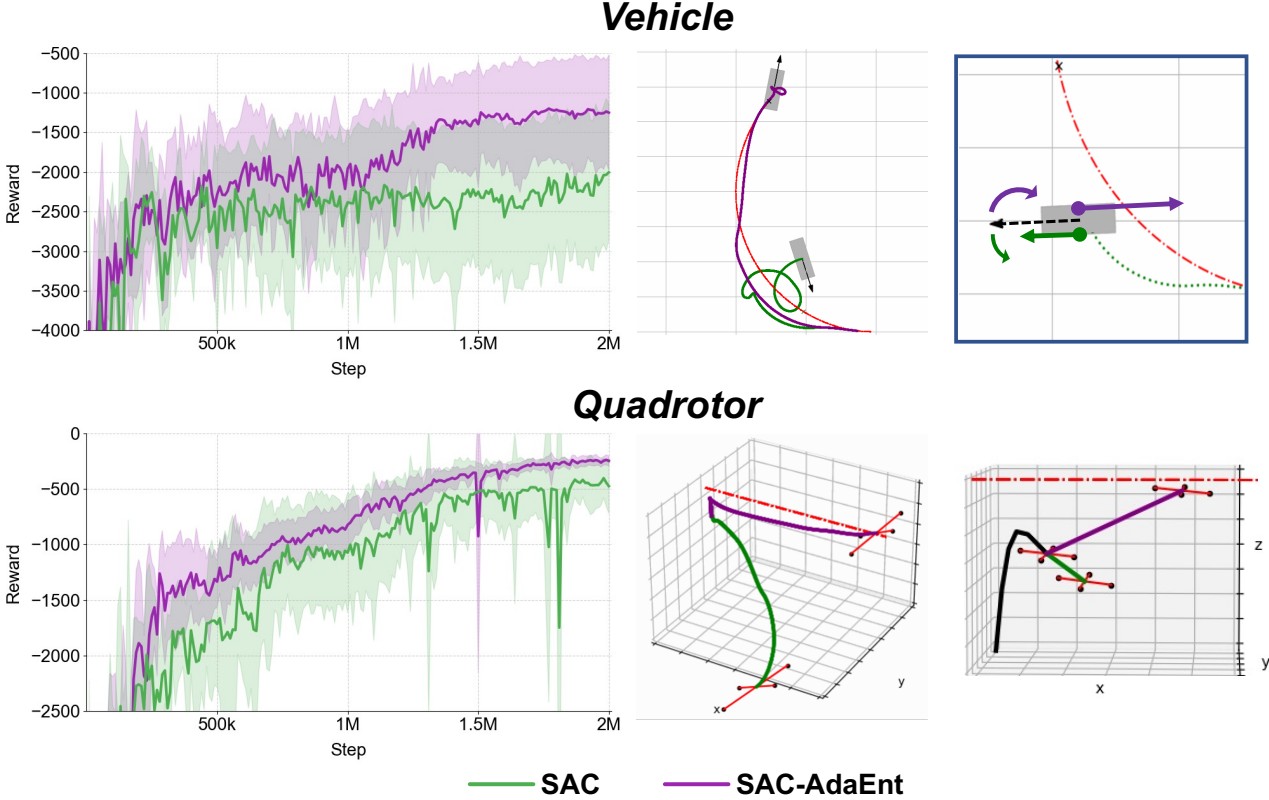

*Figure 15.* Performance of SAC-AdaEnt v.s. SAC. **Left:** Reward Improvement. **Middle:** Full trajectory rendering. **Right:** Behavior of policy on critic states. In *Vehicle*, SAC-AdaEnt successfully steers and brakes to bring the vehicle back on track, while in *Quadrotor*, it effectively lifts the quadrotor to follow the designated path.

## F. Additional Experiments on Other MaxEnt algorithm

Although SAC is a powerful MaxEnt algorithm, to ensure our findings generalize beyond SAC's particular implementation of entropy regularization, we also evaluate Soft Q-Learning (SQL), an alternative MaxEnt method. SQL extends traditional Q-learning by incorporating an entropy bonus into its Bellman backup, resulting in a policy that maximizes both expected return and action entropy—thereby fitting within the maximum-entropy RL framework. It can extend to continuous actions by parameterizing both the soft $Q$-function and policy with neural networks and using the reparameterization trick for efficient, entropy-regularized updates. We compare the performance of all algorithms on *Vehicle*, *Quadrotor* and *Hopper*. The results in Table. 4 show that SQL also suffers from the entropy-misleading issue, but its AdaEnt variant effectively mitigates this weakness.

| Algorithm | Vehicle | Quadrotor | Hopper |
|---|---|---|---|
| SAC ($\alpha = 0.2$) | $-2003.85 \pm 867.82$ | $-475.29 \pm 244.96$ | $3484.46 \pm 323.87$ |
| SAC (auto-$\alpha$) | $-1551.96 \pm 636.88$ | $-666.62 \pm 233.19$ | $2572.00 \pm 901.35$ |
| SAC-AdaEnt | $-1250.45 \pm 725.40$ | $-247.58 \pm 45.15$ | $3285.17 \pm 958.43$ |
| SQL | $-2715.48 \pm 453.00$ | $-6082.51 \pm 1632.35$ | $2998.21 \pm 158.19$ |
| SQL-AdaEnt | $-2077.59 \pm 266.84$ | $-4499.35 \pm 863.73$ | $3115.94 \pm 25.19$ |

*Table 4.* Performance (mean $\pm$ std) across Vehicle, Quadrotor, and Hopper tasks for various algorithms.

