# OpenReview forum: "When Maximum Entropy Misleads Policy Optimization"
_ICML.cc/2025/Conference — ICML 2025 poster_

### Official Review · Reviewer_xbLf · 2025-03-14

**Overall Recommendation:** 3

**Summary:**

The paper examines failure cases of Maximum Entropy (MaxEnt) RL, showing how entropy maximization can mislead policy optimization. It introduces the Entropy Bifurcation Extension, a theoretical construct proving that MaxEnt RL can drive policies toward suboptimal actions (Theorem 5.5, Proposition 5.6). Empirical results demonstrate SAC failures in vehicle control, quadruped locomotion, and drone tracking due to misleading entropy landscapes. The authors explore adaptive entropy tuning as a mitigation strategy but acknowledge its limitations.

## update after rebuttal
I appreciate the authors’ detailed response and additional experimental results. The clarification about the theoretical setting and the empirical motivation for SAC-AdaEnt helped address my main concerns. The new comparisons with Soft Q-Learning and discussions of performance across more environments (e.g., Hopper, Vehicle, Quadrotor) improved the empirical grounding of the claims.

I still believe that broader algorithmic comparisons and more extensive ablations (e.g., on entropy scaling) would further strengthen the paper, but I find the core contributions valuable. I maintain my score, and I look forward to seeing the suggested revisions incorporated into the final version.

**Claims And Evidence:**

1. The claim that entropy can mislead policy optimization is well-supported by both theoretical analysis and empirical results.
2. Theoretical constructs such as the Entropy Bifurcation Extension and its proofs are rigorously presented and demonstrate the misalignment between MaxEnt RL and true optimal policies.
3. The empirical experiments clearly show cases where SAC fails due to entropy distortion.

**Essential References Not Discussed:**

The paper does not cite Ahmed et al., "Understanding the Impact of Entropy on Policy Optimization" (ICML 2019), which analyzes how entropy smooths optimization landscapes and improves convergence. Since this submission focuses on failure cases of entropy regularization, discussing Ahmed et al. would provide a more balanced perspective on when entropy is beneficial versus when it may mislead policy optimization.

**Experimental Designs Or Analyses:**

Advantages
1. The experiments effectively demonstrate failure cases of MaxEnt RL in vehicle control, quadruped locomotion, and drone tracking tasks.
2. The empirical results support the theory, showing how SAC fails due to misleading entropy landscapes.

Limitations
1. The study focuses on SAC but does not evaluate whether other entropy-based methods (e.g., MPO, Soft Q-Learning) experience similar failures.
2. Could benefit from conducting ablation studies on entropy-related hyperparameters, such as temperature scaling, to analyze their role in failure cases.

**Methods And Evaluation Criteria:**

1. The evaluation includes diverse continuous control tasks, which are relevant benchmarks for MaxEnt RL.
2. The paper does not propose a new RL algorithm but instead provides an analysis of existing methods, which may limit its impact as a technical contribution.

**Other Comments Or Suggestions:**

1. Clarifying how adaptive entropy tuning compares to alternative exploration techniques (e.g., intrinsic motivation, Bayesian exploration) would add value.
2. A more detailed comparison between SAC and other entropy-based RL methods (e.g., MPO, Soft Q-Learning) would provide a broader picture.

**Other Strengths And Weaknesses:**

Strengths:
1. The paper provides new theoretical insights into the failure cases of MaxEnt RL.
2. The experiments are well-designed and effectively demonstrate the theoretical claims.
3. The Entropy Bifurcation Extension is a novel construct for analyzing RL policies.

Minor Issue:
1. The paper does not propose a new method, making its contribution primarily analytical rather than algorithmic.

**Questions For Authors:**

1. How does the failure mode extend to other entropy-based RL methods?
2. How sensitive is the failure mode to the choice of reward scaling and environment structure?

**Relation To Broader Scientific Literature:**

The paper contributes to the understanding of when entropy regularization fails in RL, contrasting with prior work that highlights its benefits (e.g., Ahmed et al.,2019)​. By providing a complementary perspective, it challenges the assumption that entropy always improves policy optimization and explores conditions where it can instead mislead learning.

**Theoretical Claims:**

1. The theoretical analysis correctly identifies failure modes of MaxEnt RL and proves that entropy can mislead policies into suboptimal behaviors.
2. The Entropy Bifurcation Extension (Theorem 5.5, Proposition 5.6) is a novel and valid mathematical construct that highlights the downsides of entropy maximization.
3. The theory does not offer a general solution to these failures but instead highlights when and why they occur.

---

> ### Author Rebuttal · Authors · 2025-03-30
>
> Thank you for your positive feedback and questions. Indeed, we focused on understanding the principles behind potential limitations of MaxEnt (SAC being its best performing and most widely-used version), complementing most existing results on the benefits of entropy regularization. Although we phrase the effect as "misleading", we showed that it can be either beneficial or harmful in different environments (Section 6.2 discusses its benefits). Consequently, the formulation of the algorithm SAC-AdaEnt is intended more as an ablation study mechanism to empirically test the validity of our theoretical analysis in practice, rather than as a new candidate of MaxEnt algorithms. At the same time, we definitely agree with the reviewer that evaluating the algorithm on more baselines and hyperparameter changes can strengthen the empirical understanding and contribution of the paper. Here we respond to the questions directly, and will extend our paper along these lines accordingly.
>
> **Q: (More MaxEnt baselines)**
>
> > How does the failure mode extend to other entropy-based RL methods? A more detailed comparison between SAC and other entropy-based RL methods (e.g., MPO, Soft Q-Learning) would provide a broader picture.
>
> **A:**
>
> We will follow your suggestion and expand the experiment section with more baseline MaxEnt algorithms. Here we provide results from additional experiments comparing with Soft Q-learning (SQL) and with adaptive entropy (SQL-AdaEnt). We observe that the overall performance of SQLs generally underperforms compared to SAC, and SQL is affected by the misleading effect in the Vehicle and Quadrotor environments as well, in which case SQL-AdaEnt also improves the performance.
>
> In the table below, we summarize the mean and variance of the policy return in the environments of Vehicle, Quadrotor (as examples of negative misleading effect) and Hopper (positive misleading effect). We see that SAC generally performs better than SQL across all environments, and the effect of AdaEnt tends to be similar on SQL and SAC. That is, in Vehicle and Quadrotor, where SQL and SAC struggled, the AdaEnt component led to better performance. The improved performance of SQL-AdaEnt on Hopper may be due to SQL being a relatively weak baseline.
>
> | **Algorithm**| **Vehicle** | **Quadrotor** | **Hopper** |
> |-|-|-|-|
> | SQL| -2715.48 ± 453.00  | -6082.51 ± 1632.35 | 2998.21 ± 158.19    |
> | SQL-AdaEnt | -2077.59 ± 266.84  | -4499.35 ± 863.73  | 3115.94 ± 25.19     |
> | SAC ($\alpha=0.2$)  | -2003.85 ± 867.82  | -475.29 ± 244.96   | 3484.46 ± 323.87    |
> | SAC auto-$\alpha$   | -1551.96 ± 636.88  | -666.62 ± 233.19   | 2572.00 ± 901.35    |
> | SAC-AdaEnt| -1250.45 ± 725.40  | -247.58 ± 45.15    | 3285.17 ± 958.43    |
>
> The results align with the predictions of our theoretical analysis, which showed that the misleading effect arises from the maximum entropy objective itself. As long as an algorithm aims to optimize policies to reduce divergence from the Boltzmann policy which is optimal under the MaxEnt objective, its performance can be negatively affected in situations that we analyzed in the paper.
>
> **Q: (More comparisons and ablation study)**
>
> > Clarifying how adaptive entropy tuning compares to alternative exploration techniques (e.g., intrinsic motivation, Bayesian exploration) would add value. How sensitive is the failure mode to the choice of reward scaling and environment structure? Could benefit from conducting ablation studies on entropy-related hyperparameters, such as temperature scaling, to analyze their role in failure cases.
>
> **A:**
>
> These are very important suggestions that we will follow to strengthen the empirical analysis in the paper. In general, our experience is that policy optimization methods that encourage more exploration generally underperform in control environments where low-entropy policies are the key to success, compared to conservative and more greedy approaches such as PPO. On the other hand, similar to our discussion in Section 6.2, in many environments the "misleading" effect of MaxEnt and enhanced exploration strategies can be crucial to performance. Based on the theoretical analysis framework that we proposed in this paper, we believe it is promising to develop fine-grained analysis methods that take into account of both the specific structure of an environment to inform the hyperparameter choices and exploration strategies.
>
> **Q: (Related Work)**
>
> > The paper does not cite Ahmed et al., "Understanding the Impact of Entropy on Policy Optimization" (ICML 2019)
>
> **A:**
>
> The paper was cited in the Related Work section on Line 90 as (Ahmed et al. 2019). We will add more discussion in the paper for this and similar analysis on the benefits of entropy regularization.

---

> > ### Comment · Reviewer_xbLf · 2025-04-03
> >
> > Thank you for your detailed responses. Your new empirical results and willingness to extend the paper accordingly are helpful.
> >
> > Also, I apologize for my earlier oversight regarding the citation of Ahmed et al. (2019). I acknowledge that the paper is indeed cited in your Related Work section, and I appreciate your clarification.
> >
> > I look forward to seeing these improvements reflected in the final version of the paper.

---

### Official Review · Reviewer_Rq6M · 2025-03-16

**Overall Recommendation:** 3

**Summary:**

The paper examines the trade-off in MaxEnt RL: while the entropy term fosters exploration and robustness, it can mislead policy optimization in tasks requiring precise, low-entropy actions, often causing MaxEnt methods like SAC to converge to suboptimal policies.

The authors formalize this phenomenon with a toy example and introduce the Entropy Bifurcation Extension, showing that auxiliary states can create misleading soft Q-value landscapes that divert the MaxEnt-optimal policy from the true optimal one.

To address this, the paper proposes SAC-AdaEnt, an adaptive tuning mechanism that monitors discrepancies between soft and plain Q-values, dynamically shifting policy updates to rely more on plain Q-values when needed.

**Claims And Evidence:**

Yes.

**Essential References Not Discussed:**

No.

**Experimental Designs Or Analyses:**

I examined the experimental designs and analyses in the paper, particularly those illustrated in Figures 4–11, which involve continuous control tasks such as Vehicle control, Quadrotor control, and standard benchmarks like Acrobot, Obstacle2D, and Hopper.

The paper compares SAC (with automated entropy adjustment) against PPO and the proposed SAC-AdaEnt. This setup is generally appropriate to highlight the effect of entropy on policy optimization. However, the choice of baselines is somewhat narrow. Including additional variants or state-of-the-art Max-Ent algorithms could have provided a more comprehensive comparison.

**Methods And Evaluation Criteria:**

Yes,

**Other Comments Or Suggestions:**

See Questions.

**Other Strengths And Weaknesses:**

While the adaptive mechanism (SAC-AdaEnt) is promising, the paper itself notes that measuring discrepancies between soft and plain Q-value landscapes requires a “global” understanding at each state. In high-dimensional environments, this might become computationally expensive or infeasible.

**Questions For Authors:**

1. The paper does not specifically emphasize that PPO also uses an entropy term. While PPO does incorporate an entropy bonus to encourage exploration, this is treated as a secondary regularizer rather than a core objective. In contrast, the MaxEnt framework—exemplified by SAC—integrates entropy maximization directly into the learning objective, which can lead to the misleading effects discussed in the paper. This distinction is critical, as it highlights why PPO may avoid some of the pitfalls of overemphasizing entropy, thereby achieving more precise control in performance-critical tasks.
2. In Figure 11， the SAC-auto-alpha refers to the popular automated entropy adjustment trick or the AdaEnt mechanism proposed in this paper? Could you compare the SAC-AdaEnt to the SAC-automated-alpha-tuning?
3. The paper introduces an adaptive entropy tuning mechanism, SAC-AdaEnt, which effectively mitigates the misleading effects of high entropy in the SAC framework. However, it is noteworthy that the proposed tuning mechanism is applied exclusively to SAC. It would enhance the paper if the authors could discuss the potential for extending this adaptive mechanism to other MaxEnt-based algorithms. Such a discussion could broaden the applicability of their approach and provide insights into whether similar benefits could be realized across a wider range of RL methods.



---

**Updated Review:** Thanks for your response — I think the problem you’re tackling is important, and the added experiments are helpful. I remain positive about the paper. Just a suggestion: it would be stronger if the main comparison focused on SAC with and without the entropy term. I know you included this in Figure 6, but putting it earlier might help clarify the main point. Since SAC and PPO differ in many ways, it’s hard to tell if the gap in Figure 1 is really due to entropy.

**Relation To Broader Scientific Literature:**

The paper builds on and extends several streams in the RL literature - MaxEnt RL, PPO, exploration versus exploitation.

**Theoretical Claims:**

Yes.

---

> ### Author Rebuttal · Authors · 2025-03-30
>
> Thank you for your positive feedback and questions. We agree with the reviewer that comparisons with wider range of baselines will enhance the empirical analysis, and we will expand the paper accordingly. Our current formulation of SAC-AdaEnt is mainly intended for empirically testing the theoretical analysis, to show how the misleading effect practically affects policy optimization (in both negative and positive ways). As our theoretical analysis shows, the effect is rooted in the use of the MaxEnt objective, and thus we believe the principles are applicable to MaxEnt algorithms in general. We discuss additional experiments with Soft Q-Learning below as an example that reflects this.
>
> **Q: (PPO with entropy)**
>
> > (Entropy in PPO) is treated as a secondary regularizer rather than a core objective. This distinction highlights why PPO may avoid some of the pitfalls of overemphasizing entropy.
>
> **A:**
>
> Thank you for pointing this out: it exactly aligns with our understanding. Our core construction of entropic bifurcation relies on the fact that the objective of policy optimization in MaxEnt algorithms is to match the Boltzmann distribution, which fundamentally changes the goal of the optimal policy and enables the misleading effect of entropy. In policy gradient methods using entropy as exploration strategies, the objective for policy optimization is unaffected, and thus entropy regularization does not lead to misleading effects as how we formulated in the paper.
>
> **Q: (SAC-auto-alpha v.s. SAC-AdaEnt)**
>
> > In Fig.11, the SAC-auto-alpha refers to automated adjustment? Could you compare the SAC-AdaEnt to the SAC-automated-alpha-tuning?
>
> **A:**
>
> Yes, in Fig.11, SAC-auto-alpha refers to automated entropy adjustment. We show more results in the next question, and will update the plots in the paper along with AdaEnt applied to other MaxEnt baselines, as you suggested. Note that SAC-auto-alpha applies a uniform entropy adjustment across all states while SAC-AdaEnt adapts the entropy individually per state.
>
> **Q: (Baselines)**
> > The choice of baselines is narrow; It would enhance the paper if extending the adaptive mechanism to other MaxEnt algorithms."
>
> **A:**
>
> We will follow the reviewer's suggestion and expand the experiment section with more baseline MaxEnt algorithms. Here we provide results from additional experiments comparing with Soft Q-learning (SQL) and with adaptive entropy (SQL-AdaEnt). We observe that the overall performance of SQLs generally underperforms compared to SAC, and SQL is affected by the misleading effect in the Vehicle and Quadrotor environments as well, in which case SQL-AdaEnt also improves the performance.
>
> In the table below, we summarize the mean and variance of the policy return across Vehicle, Quadrotor (as examples of negative misleading effect) and Hopper (positive misleading effect). We see that SAC generally performs better than SQL across all environments, and the effect of AdaEnt is similar on SQL and SAC. That is, in Vehicle and Quadrotor, where SAC struggled, the AdaEnt component led to better performance. The improved performance of SQL-AdaEnt on Hopper may be due to SQL being a relatively weak baseline.
>
> | **Algorithm**| **Vehicle** | **Quadrotor** | **Hopper** |
> |-|-|-|-|
> | SAC ($\alpha=0.2$)  | -2003.85 ± 867.82  | -475.29 ± 244.96   | 3484.46 ± 323.87    |
> | SAC auto-$\alpha$   | -1551.96 ± 636.88  | -666.62 ± 233.19   | 2572.00 ± 901.35    |
> | SAC-AdaEnt| -1250.45 ± 725.40  | -247.58 ± 45.15    | 3285.17 ± 958.43    |
> | SQL| -2715.48 ± 453.00  | -6082.51 ± 1632.35 | 2998.21 ± 158.19    |
> | SQL-AdaEnt | -2077.59 ± 266.84  | -4499.35 ± 863.73  | 3115.94 ± 25.19     |
>
> The results align with the predictions of our theoretical analysis, which showed that the misleading effect arises from the MaxEnt objective itself. As long as an algorithm aims to reduce divergence from the Boltzmann policy, the performance can be negatively affected as analyzed in the paper.
>
> **Q: (Scalability)**
> > While the adaptive mechanism is promising, measuring discrepancies requires a “global” understanding at each state. In high-dimensional environments, ... computationally expensive.
>
> **A:**
>
> Yes, we formulated SAC-AdaEnt to test the theory in practice, acknowledging that it is not intended as a scalable replacement for SAC. As we shown in Section 6.2, the misleading effects of entropy may explain some of the key benefits of SAC, and thus we do not always need to eliminate the effect. We believe a promising direction in challenging environments is to use baseline SAC and PPO algorithms to understand the gap between Q-plain and Q-soft landscapes first, and then modulate the use of entropy on specific regions in the state space to minimize the need for global probing. Consequently, we focused mainly on analyzing the principles of how entropy affects policy optimization, in the hope that the understanding leads to more nuanced policy optimization strategies in challenging environments.

---

### Official Review · Reviewer_gm3Q · 2025-03-17

**Overall Recommendation:** 4

**Summary:**

The authors analyze the trade-off between exploration/robustness and exploitation in Maximum Entropy Reinforcement Learning though a variety of control tasks. The paper demonstrate that in performance-critical control tasks requiring precise, low-entropy actions, Maximum Entropy approaches can become misguided by their entropy-driven exploration, which help readers to understand how entropy maximization affects policy optimization.

**Claims And Evidence:**

Yes, main claims made in the submission are generally supported by clear evidence.

**Essential References Not Discussed:**

NA

**Experimental Designs Or Analyses:**

The authors demonstrate empirically with a toy example and experiments on complex control tasks to show explicit situations where maximizing entropy leads to convergence towards suboptimal, high-entropy solutions, which I found is very clear and sound.
Issues: the authors also propose a possible solution, SAC-AdaEnt. SAC-AdaEnt is then compared with standard SAC, however, limited discussion/analysis is on SAC-AdaEnt, for example, will SAC-AdaEnt decreases performance in tasks where SAC generally succeed (e.g. Hopper)?

**Methods And Evaluation Criteria:**

The analysis of the misleading effect of maximum entropy RL is conducted on different robotics control environments with 5 random seeds. Results are compared to PPO to highlight performance gaps and benefits of entropy maximization at different situations. Authors also proposed SAC with Adative Entropy Scaling, SAC-AdaEnt to further show the negative effective of soft Q values in some situations.

**Other Comments Or Suggestions:**

See above sections

**Other Strengths And Weaknesses:**

See above sections

**Questions For Authors:**

Were entropy-traps also expected or observed in high-dimensional image-based RL tasks?
Will SAC-AdaEnt decreases performance in tasks where SAC generally succeed (e.g. Hopper, or other Mujoco envs)?

**Relation To Broader Scientific Literature:**

The paper is connected to important literature areas of maximum entropy RL and exploration-exploitation trade-off.
By clearly illustrating situations where maximizing entropy can degrade performance, this paper significantly extends the understanding of limitations inherent in widely used maximum entropy methods such as SAC.
This paper also explicitly compares non-maximum entropy algorithm PPO against SAC, highlighting trade-offs between entropy-driven exploration and precise exploitation on practical tasks.
With these discussions, I think the paper can help people improve current maximum-entropy algorithms int he future.

**Theoretical Claims:**

Theoretical part looks ok to me.

---

> ### Author Rebuttal · Authors · 2025-03-30
>
> Thank you for your positive feedback and questions. We agree that more discussion and analysis can be done for SAC-AdaEnt, and we provide some more details below. Indeed, we formulated SAC-MaxEnt less as an algorithm that is intended to replace SAC, but more as an ablation mechanism that tests the validity of the theoretical analysis in practical learning environments. We believe the principled understanding of the misleading effect of MaxEnt algorithms (and how it can both positively and negatively affect policy learning) opens up the possibility of developing more customized policy optimization procedures for challenging environments that exploit their specific structures and value landscapes.
>
> **Q: (SAC-AdaEnt)**
>
> > Will SAC-AdaEnt decrease performance in tasks where SAC generally succeeds?
>
> **A:**
>
> Following the question, we performed additional experiments on the environments in the paper where SAC generally succeeds: Hopper, Obstacle2D, Acrobot (also attached results for Vehicle and Quadrotor, in which SAC fails). The summary of performance comparison is shown in the table below, and we will add the plots and more discussion in the paper.
>
> We observe that SAC-AdaEnt led to less mean return compared SAC as well as higher variance. Note that our design of SAC-AdaEnt maintains the entropy-based exploration component in SAC (only changing the policy optimization objective). Thus the overall performance still benefits from exploration, and the higher variance is the result of the advantage landscape being less dominated by entropy. The obstacle2D and acrobot environments are lower-dimensional, and thus exploration itself is likely sufficient for discovering successful trajectories, and thus the performance has not degraded in SAC-AdaEnt.
>
>
> | **Algorithm**       | **Hopper**          | **Obstacle2D**      | **Acrobot**       | **Vehicle**        | **Quadrotor**      |
> |---------------------|--------------------|--------------------|---------------------|---------------------|---------------------|
> | SAC   | 3484.46 ± 323.87    | 501.98 ± 0.62       | -45.25 ± 7.94       | -2003.85 ± 867.82  | -475.29 ± 244.96   |
> | SAC-AdaEnt          | 3285.17 ± 958.43     | 501.50 ± 0.57       | -36.31 ± 16.42      | -1250.45 ± 725.40  | -247.58 ± 45.15   |
>
>
> Overall, we believe the experiments show that in environments where SAC performs well, SAC-AdaEnt can still rely on entropy to support exploration, while the performance may benefit less from the tendency of the standard SAC in escaping action regions with raw high Q-values.
>
> **Q: (High-dimensional RL)**
> > Were entropy-traps also expected or observed in high-dimensional image-based RL tasks?
>
> **A:**
>
> Entropy-traps are fundamentally dependent on the reward structure and the environment dynamics in the MDP, in terms of whether the success of learning relies on low-entropy policies. Thus the misleading effects can definitely be observed in high-dimensional RL problems. We believe the key to observing entropy traps may not always be about the dimensionality of the state space, but more crucially determined by the dimensionality of the action space -- when the action space is high-dimensional, it is more likely that the feasible policies form a low-entropy distribution over the space, thus making it easier to observe the misleading effect of MaxEnt. We believe such problems can arise frequently in high-dimensional RL problems in general, such as fine-tuning of language models, foundation Vision-Language-Action models, as well as diffusion policies.

---

### Official Review · Reviewer_rvMR · 2025-03-17

**Overall Recommendation:** 3

**Summary:**

This paper posits that the entropy maximization objective in SAC can lead to failure in tasks that require precise, low-entropy policies, essentially "misleading" policy optimization.

**Claims And Evidence:**

No. There are several problematic claims.

1. The central claim about the misleading effect relies on $\alpha = 1$, whereas it is well-known that the entropy coefficient needs to be tuned according to the environment.

L627 says "without loss of generality, we use $\alpha = 1$ for the entropy coefficient", but this assumption does break generality. Naturally, with a high value of $\alpha$, SAC is not expected to work, so this "theoretical" calculation for the toy example does not do anything meaningful.

Crucially, for the special case of $\alpha = 0$, the MaxEnt effect of SAC would turn off, and the optimal policy would not suffer from the misleading effect. So, trivially, one cannot assume a value of $\alpha$ without losing generality.

In fact, the entire argument of the paper relies on a large enough value of $\alpha$. If $\alpha$ was not large, then there would not be any misleading effect at all.

2. "However, MaxEnt methods have also been shown to struggle with performance-critical control problems in practice,
where non-MaxEnt algorithms can successfully
learn" is a very vague claim in the abstract, and the works cited in this paper do not give a conclusive evidence of MaxEnt algorithms always struggling in important control problems. As a matter of fact, SAC is indeed applied to many useful real-world control problems.

3. "When a more realistic dynamics model for the quadrotor is used, then SAC always fails, while PPO can succeed under the same initialization and dynamics." — the term more realistic is vague and hand-wavy.

4. "In both the Vehicle and Quadrotor environments, the policy learned by SAC-AdaEnt mostly corrects the behavior of the SAC policy, as illustrated in their overall trajectories and the critical shown in the plots." is not a valid claim given the negligible improvement shown with AdaEnt in Figure 9.

**Essential References Not Discussed:**

N/A

**Experimental Designs Or Analyses:**

- The experiments only analyze role of SAC at a fixed $\alpha$, which does not really represent the importance or drawbacks of the entropy term in SAC, because the entropy coefficient plays a key part.
- The comparison between SAC and PPO is not exactly fair because one is off-policy and the other is on-policy.

**Methods And Evaluation Criteria:**

- Method: No new method is proposed and without understanding the role of $\alpha$, the SAC and PPO methods tested are not enough to understand the claims behind the role of entropy.
- The selected benchmarks seem to be arbitrarily selected and "complex dynamics" is used without any justification of what is complex.

**Other Comments Or Suggestions:**

N/A

**Other Strengths And Weaknesses:**

The central claim that high entropy regularization leads to poor policy optimization is trivial, which SAC already solves by having a tunable entropy coefficient $\alpha$. The analysis with a fixed high value of $\alpha$ in this paper does not offer any useful insights.

**Questions For Authors:**

N/A

**Relation To Broader Scientific Literature:**

Many claims are not justified well, and papers like Tan & Karakose (2023) are cited to claim that SAC delivers suboptimal solutions in comparison to PPO in complex control problems, which is not shown in those papers.

**Theoretical Claims:**

The analysis in Appendix A is meaningless if the authors assume $\alpha=1$ because they assume a fixed and large value of $\alpha$. Naturally, if one changes the RL objective from reward optimization too much, then the policy optimization would fail at convergence.

---

> ### Author Rebuttal · Authors · 2025-03-30
>
> Thank you for your feedback and questions. We will reorganize the writing to make the core claims clear from the beginning of the paper. The toy example seems to have obscured the core results, which we further explain here.
>
> **Q: Large $\alpha$ value?**
>
> > - In fact, the entire argument of the paper relies on a large enough value of $\alpha$. If $\alpha$ was not large, then there would not be any misleading effect at all.
> >
> > - The analysis in Appendix A is meaningless if the authors assume because they assume a fixed and large value of alpha.
> >
> > - SAC already solves by having a tunable entropy coefficient. The analysis with a fixed high value does not offer any useful insights.
>
> **A:**
>
> We do not assume large fixed $\alpha$ values in the core results.
>
> We agree with the reviewer that under a large $\alpha$ value it seems obvious that the learning problem is changed. This is why in Section 3 we derive the general result: the misleading effect can be shown with *arbitrarily small* positive $\alpha$. All theorems in Section 5 treat $\alpha$ as a variable in the definitions and proofs to construct the bifurcation extension for misleading effects.
>
> To see how: for an arbitrary $\alpha$, the optimal MaxEnt policy follows $\pi^*(a|s)=\exp(\alpha^{-1}Q(s,a))/Z(s))$ with normalizing factor $Z(s)=\int \exp(\alpha^{-1}Q(s,a))da$ (see [Haarnoja, 2018b] Eq (4)). The Q terms are thus canceled out in the state values:
> \begin{align*}
> V(s)&=\mathbb{E}_{a\sim\pi^*}[Q(s,a)-\alpha \log(\pi^*(\cdot|s))]\\\\
> &=\mathbb{E}_a[\cancel{Q(s,a)}-\alpha\cdot\bigg(\cancel{\log(\exp}(\alpha^{-1}\cancel{Q(s,a)}))-\log Z\bigg)] \\\\
> &=\mathbb{E}_a[\alpha \log Z]\\\\
> &=\alpha \log(Z)
> \end{align*}
> Consequently, the misleading effect preserves as long as the value ordering between good and bad states incorporates the scaling effect of $\alpha$. Moreover, with our definition of the bifurcation extension, if MaxEnt is misled for any small $\alpha$, then with any $\alpha'\geq \alpha$ the MaxEnt is misled as well. Thus the results apply to auto-tuning by choosing $\alpha$ to be smaller than the final weight after the decay schedule in SAC with autotuning.
>
> Specifically, the toy example is designed to allow the use of any positive $\alpha$. For instance, say $\alpha=0.00001$, then we would multiply with $\alpha^{-1}$ in the exponents on Line 638 and 642 in Appendix A.1.1. Thus, to maintain the misleading effect we can scale all rewards by $\alpha$ to cancel out $\alpha^{-1}$, which preserves the same integrals and the ordering of $\log Z(s_g)<\log Z(s_b)$. Thus, in the example $Q(s_0,a\in A_1)=\gamma V(s_g)=\gamma\alpha \log(Z(s_g))<\gamma\alpha \log(Z(s_b))=Q(s_0,a\in A_2)$. Namely, the MaxEnt-optimal policy still prefers the wrong action range $a\in A_2$. Note that this scaling is why in theoretical analysis of MaxEnt the temperature $\alpha$ is typically omitted (original SAC paper [Haarnoja, 2018a], remark after Eq(1)).
>
> All experiments in Section 6 are run with SAC with small $\alpha$ and autotuning (Line 923 Table 2).
>
> We will update the paper to emphasize these points. Thank you for pointing out the concern.
>
> **Q: Complex Dynamics?**
>
> > - When a more realistic dynamics model for the quadrotor is used... — the term more realistic is vague and hand-wavy.
> > - The selected benchmarks seem to be arbitrarily selected and complex dynamics is used without any justification of what is complex.
>
> **A:**
>
> The difference in the dynamics of quadcopters between easy and complex is described in Appendix C.2 and we will further add the full equations. We agree with the reviewer that it is hard to quantify complex dynamics, so we should refer to them as environments that require low-entropy control policies to succeed, which can be observed more directly by plotting the advantage landscapes as in Section 6.1.
>
> The benchmarks are selected for visualizing concretely *when* entropy affects policy updates, as shown in Section 6. We also showed that the misleading effect can also benefit learning (Section 6.2). Our goal is to understand when MaxEnt helps and when it hinders policy optimization.
>
>
> **Q: No performance issues in SAC?**
>
> > - The works cited in this paper do not give conclusive evidence of MaxEnt algorithms always struggling in important control problems.
> > - Tan \& Karakose (2023) are cited to claim that SAC delivers suboptimal solutions in comparison to PPO in complex control problems, which is not shown in those papers.
>
> **A:**
>
> We do not aim to show that MaxEnt algorithm *always* struggles, but to understand when the entropy terms *may* hinder learning.
>
> We definitely agree that SAC is widely observed to outperform policy gradient, which makes it important to understand in what cases they may underperform. The cited practical robotics papers are examples of such cases. In particular, Tan \& Karakose (2023) shows in Figure 6 that SAC is outperformed by other methods. We can remove the reference if it is not a convincing example.

---

> > ### Comment · Reviewer_rvMR · 2025-04-04
> >
> > Thank you for your response. It is still unclear to me how the analysis in this paper that compares SAC v/s PPO clearly disentangles the performance deficit in Vehicle, Quadrotor, and OpenCat to the fact that SAC has entropy regularization. In fact, there are easy ways to test this:
> > 1. DDPG should outperform SAC in these environments, because DDPG is not misled by entropy. However, as shown in Figure 11, there is no performance difference and PPO >> DDPG ~ SAC. So, clearly the difference in SAC and PPO cannot be attributed to the presence of entropy.
> > 2. PPO with entropy should outperform PPO without entropy. As Table 1 states, the PPO experiments are with entropy coefficient = 0. However, one could easily experiment with larger entropy coefficients (like 0.2) and analyze the Q-function curves learned by PPO. While PPO does not learn a "soft" Q-function, this objective would still lead to a high entropy actor which would fail in tasks that require precise, low-entropy policies — as claimed in this paper.
> >
> > Overall, I don't find the experiments convincing to show that the real reason for the performance difference in these environments is, in fact, due to the presence of entropy regularization in SAC, as opposed to other differences between the algorithms. I understand and appreciate that entropy-regularized learning in SAC would lead to a final policy that must balance between the environment reward and entropy maximization, which means it cannot be fully optimal — both intuitively and theoretically, I agree this is true. But, whether this difference has an impact on the environments considered in this paper is not justified.

---

> > > ### Author Response · Authors · 2025-04-06
> > >
> > > Thank you for your response and additional questions.
> > >
> > > > I understand and appreciate that entropy-regularized learning in SAC would lead to a final policy that ... cannot be fully optimal both intuitively and theoretically, I agree this is true.
> > >
> > > Thank you for reading our rebuttal. We are glad that it clarified your previous questions, and that our main results are now clear.
> > >
> > > We now provide explanations on the experiments.
> > >
> > > > I don't find the experiments convincing to show that the real reason for the performance difference in these environments is, in fact, due to the presence of entropy regularization in SAC, as opposed to other differences between the algorithms.
> > >
> > > We completely agree, which is why the experiments are not used to explain the differences between these RL algorithms.
> > >
> > > Instead, we focused on evaluating the practical relevance of our theoretical claims, by answering the following questions:
> > >
> > > **(A)** In practice, can MaxEnt produce value landscapes that mislead policy optimization?
> > >
> > > **(B)** If so, how does using vs. not using MaxEnt objectives affect the behavior of SAC in practice, assuming all other components of the algorithm are fixed?
> > >
> > > We addressed (A) in Section 6.1 by showing that in several control environments, SAC generated soft-Q landscapes that misled policy centers to actions that lead to failure (Figure 5 and 6).
> > >
> > > We addressed (B) in Section 6.3 by formulating SAC-AdaEnt which only switches out soft-Q values when the discrepancy between soft-Q and plain-Q values is large. By doing so, the problematic action choices shown in (A) were corrected (Figure 9).
> > >
> > > The reason that we compare with PPO is not to explain its difference with SAC. Instead, given the fact that PPO performs well on these environments, it gives us good action choices that can successfully control the agents. Then by plotting the actions from PPO and SAC on the soft-Q value landscapes, we confirm the misleading effects.
> > >
> > > In the main text (Figure 4), we have not compared with DDPG or other algorithms, again because the goal is not to compare performances. The learning curves are used to confirm the overall impact of critical action choices, since Figure 5 and 6 can only show concrete states.
> > >
> > > We respond to your two specific questions as follows.
> > >
> > > > 1. DDPG should outperform SAC in these environments, because DDPG is not misled by entropy... So, clearly the difference in SAC and PPO cannot be attributed to the presence of entropy.
> > >
> > > DDPG and SAC differ substantially. DDPG uses a deterministic actor policy, explores by adding noise rather than sampling from policy distributions, uses a single Q-network, and updates the policy using raw Q-network gradients. SAC improves DDPG systematically using the MaxEnt framework that typically results in superior performance, leading to general acceptance of the effectiveness of SAC/MaxEnt.
> > >
> > > Our work aims to alert that *while entropy offers major benefits, it is not without flaws*, complementing existing work to better understand the mechanisms behind its potential limitations. This is not an argument that the drawbacks of entropy outweigh its advantages.
> > >
> > > We do not claim that entropy’s misleading effects are the sole reason that policy optimization may struggle in challenging environments, nor do we advocate reverting to DDPG or removing entropy from SAC altogether.
> > >
> > > In fact, we have hypotheses about why DDPG may perform poorly in these environments, despite not being affected by entropy. While entropy may be misleading, it smoothes the value landscape, helping align Q-network gradients with policy improvement. In contrast, DDPG relies on raw input gradients from highly non-convex Q-networks, which can misguide policy optimization even more erratically. This issue is beyond the scope of our paper, but we could add a remark for clarity.
> > >
> > > > 2. PPO with entropy should outperform PPO without entropy. While PPO does not learn a "soft" Q-function, this objective would still lead to a high entropy actor which would fail in tasks that require precise, low-entropy policies — as claimed in this paper.
> > >
> > > We are happy to see the reviewer applying our intuition and results to this setting. As discussed, our technical results address the specific use of entropy in the MaxEnt objectives. Analyzing entropy in exploration requires different methods: as you (and Reviewer Rq6M) noted, it does not distort the Q-landscape but still hinders concentration on low-entropy policies because of exploration, rather than misleading at convergence like MaxEnt.
> > >
> > > > But, whether this difference has an impact on the environments considered in this paper is not justified.
> > >
> > > We hope the clarifications above help with a re-examination of our experiments. To summarize, we did not intend to explain the differences between SAC and other algorithms, but to evaluate the practical relevance of our theoretical analysis, by addressing (A) and (B) above through Figures 5 to 9 and the SAC-AdaEnt algorithm.

---

### Decision · Program_Chairs · 2025-05-01

**Decision:**

Accept (poster)

**Comment:**

All reviewers agree that the paper addresses a highly relevant challenge in reinforcement learning: improving our understanding of entropy regularization in relation to exploration, robustness, and the optimization landscape. They also recognize the proposed "Entropic Bifurcation Extension" as a novel and valuable contribution. The correctness of the proofs was also checked, including the appendix to a large extent.

After the rebuttal and discussion phases, there was broad consensus among the reviewers (despite one differing view) that the main claims of the paper are supported by clear and convincing evidence.

Three reviewers raised a concern that I share. It can be summarized as an apparent misalignment between the stated goals of the paper and its actual focus. Specifically, the authors place strong emphasis on comparing SAC versus PPO from the beginning, which raises the following questions: (1) why not center the analysis more directly on the presence or absence of entropy regularization, regardless of the specific algorithmic choice? and (2) why exclude other algorithms, e.g., MPO or Soft Q-Learning as examples of regularized methods, or  e.g., DDPG, as an unregularized one?

The rebuttal helped clarify misunderstandings related to this issue, as reflected in the updated reviews.
Overall, this is a well-written paper, and its contribution remains significant and valuable. I recommend acceptance.
I strongly encourage the authors to incorporate the following improvements in the revised version:

- Give greater emphasis to the comparison between SAC with and without the entropy term, rather than presenting it as a subsection of the experiments.
- Include the comparison with Soft Q-Learning and the additional discussions of performance across more environments (e.g., Hopper, Vehicle, Quadrotor) that were provided during the rebuttal phase.
- Re-check all references in the introduction. For example, Tan & Karakose (2023) is not appropriate. Also, two of the three references cited after the sentence "Often the performance by SAC is indeed shown to be inferior to PPO despite efforts in tuning" do not support the claim being made.